# Emergent Abilities of Large Language Models

**Jason Wei**[1]  *jasonwei@google.com*
**Yi Tay**[1]  *yitay@google.com*
**Rishi Bommasani**[2]  *nlprishi@stanford.edu*
**Colin Raffel**[3]  *craffel@gmail.com*
**Barret Zoph**[1]  *barretzoph@google.com*
**Sebastian Borgeaud**[4]  *sborgeaud@deepmind.com*
**Dani Yogatama**[4]  *dyogatama@deepmind.com*
**Maarten Bosma**[1]  *bosma@google.com*
**Denny Zhou**[1]  *dennyzhou@google.com*
**Donald Metzler**[1]  *metzler@google.com*
**Ed H. Chi**[1]  *edchi@google.com*
**Tatsunori Hashimoto**[2]  *thashim@stanford.edu*
**Oriol Vinyals**[4]  *vinyals@deepmind.com*
**Percy Liang**[2]  *pliang@stanford.edu*
**Jeff Dean**[1]  *jeff@google.com*
**William Fedus**[1]  *liamfedus@google.com*

[1] *Google Research*  [2] *Stanford University*  [3] *UNC Chapel Hill*  [4] *DeepMind*

**Reviewed on OpenReview:** *https://openreview.net/forum?id=yzkSU5zdwD*

## Abstract

Scaling up language models has been shown to predictably improve performance and sample efficiency on a wide range of downstream tasks. This paper instead discusses an unpredictable phenomenon that we refer to as *emergent abilities* of large language models. We consider an ability to be emergent if it is not present in smaller models but is present in larger models. Thus, emergent abilities cannot be predicted simply by extrapolating the performance of smaller models. The existence of such emergence raises the question of whether additional scaling could potentially further expand the range of capabilities of language models.

## 1 Introduction

Language models have revolutionized natural language processing (NLP) in recent years. It is now well-known that increasing the scale of language models (e.g., training compute, model parameters, etc.) can lead to better performance and sample efficiency on a range of downstream NLP tasks (Devlin et al., 2019; Brown et al., 2020, *inter alia*). In many cases, the effect of scale on performance can often be methodologically predicted via scaling laws—for example, scaling curves for cross-entropy loss have been shown to empirically span more than seven orders of magnitude (Kaplan et al., 2020; Hoffmann et al., 2022). On the other hand, performance for certain downstream tasks counterintuitively does not appear to continuously improve as a function of scale, and such tasks cannot be predicted ahead of time (Ganguli et al., 2022).

In this paper, we will discuss the unpredictable phenomena of *emergent abilities* of large language models. Emergence as an idea has been long discussed in domains such as physics, biology, and computer science (Anderson, 1972; Hwang et al., 2012; Forrest, 1990; Corradini & O'Connor, 2010; Harper & Lewis, 2012, *inter*

*alia*). We will consider the following general definition of emergence, adapted from Steinhardt (2022) and rooted in a 1972 essay called "More Is Different" by Nobel prize-winning physicist Philip Anderson (Anderson, 1972):

> *Emergence is when quantitative changes in a system result in qualitative changes in behavior.*

Here we will explore emergence with respect to model scale, as measured by training compute and number of model parameters. Specifically, we define *emergent abilities of large language models* as abilities that are not present in smaller-scale models but are present in large-scale models; thus they cannot be predicted by simply extrapolating the performance improvements on smaller-scale models (§2).[1] We survey emergent abilities as observed in a range of prior work, categorizing them in settings such as few-shot prompting (§3) and augmented prompting strategies (§4). Emergence motivates future research on why such abilities are acquired and whether more scaling will lead to further emergent abilities, which we highlight as important questions for the field (§5).

## 2 Emergent Abilities Definition

As a broad concept, emergence is often used informally and can be reasonably interpreted in many different ways. In this paper, we will consider a focused definition of emergent abilities of large language models:

> *An ability is emergent if it is not present in smaller models but is present in larger models.*

Emergent abilities would not have been directly predicted by extrapolating a scaling law (i.e. consistent performance improvements) from small-scale models. When visualized via a scaling curve ($x$-axis: model scale, $y$-axis: performance), emergent abilities show a clear pattern—performance is near-random until a certain critical threshold of scale is reached, after which performance increases to substantially above random. This qualitative change is also known as a *phase transition*—a dramatic change in overall behavior that would not have been foreseen by examining smaller-scale systems (Huberman & Hogg, 1987).

Today's language models have been scaled primarily along three factors: amount of computation, number of model parameters, and training dataset size (Kaplan et al., 2020; Hoffmann et al., 2022). In this paper, we will analyze scaling curves by plotting the performance of different models where training compute for each model is measured in FLOPs on the $x$-axis (Hoffmann et al., 2022). Because language models trained with more compute tend to also have more parameters, we additionally show plots with number of model parameters as the $x$-axis in Appendix D (see Figure 11 and Figure 12, as well as Figure 4 and Figure 10). Using training FLOPs or model parameters as the $x$-axis produces curves with similar shapes due to the fact that most dense Transformer language model families have scaled training compute roughly proportionally with model parameters (Kaplan et al., 2020).

Training dataset size is also an important factor, but we do not plot capabilities against it because many language model families use a fixed number of training examples for all model sizes (Brown et al., 2020; Rae et al., 2021; Chowdhery et al., 2022). Although we focus on training computation and model size here, there is not a single proxy that adequately captures all aspects of scale. For example, Chinchilla (Hoffmann et al., 2022) has one-fourth as many parameters as Gopher (Rae et al., 2021) but uses similar training compute; and sparse mixture-of-expert models have more parameters per training/inference compute than dense models (Fedus et al., 2021; Du et al., 2021). Overall, it may be wise to view emergence as a function of many correlated variables. For example, later in Figure 4 we will also plot emergence as a function of WikiText103 perplexity (Merity et al., 2016), which happens to closely correlate with training computation for Gopher/ Chinchilla (though this correlation may not hold in the long-run).

Note that the scale at which an ability is first observed to emerge depends on a number of factors and is not an immutable property of the ability. For instance, emergence may occur with less training compute

---

[1]This survey focuses on pre-trained Transformer language models. Emergent abilities in NLP more broadly, however, could go back to Miller et al. (2004), Liang (2005), or earlier.

or fewer model parameters for models trained on higher-quality data. Conversely, emergent abilities also crucially depend on other factors such as not being limited by the amount of data, its quality, or the number of parameters in the model. Today's language models are likely not trained optimally (Hoffmann et al., 2022), and our understanding of how to best train models will evolve over time. Our goal in this paper is not to characterize or claim that a specific scale is required to observe emergent abilities, but rather, we aim to discuss examples of emergent behavior in prior work.

## 3 Few-Shot Prompted Tasks

We first discuss emergent abilities in the *prompting* paradigm, as popularized by GPT-3 (Brown et al., 2020).[2] In prompting, a pre-trained language model is given a prompt (e.g. a natural language instruction) of a task and completes the response without any further training or gradient updates to its parameters. Brown et al. (2020) proposed *few-shot prompting*, which includes a few input-output examples in the model's context (input) as a preamble before asking the model to perform the task for an unseen inference-time example. An example prompt is shown in Figure 1.

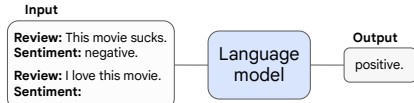

Figure 1: Example of an input and output for few-shot prompting.

The ability to perform a task via few-shot prompting is emergent when a model has random performance until a certain scale, after which performance increases to well-above random. Figure 2 shows eight such emergent abilities spanning five language model families from various work.

**BIG-Bench.** Figure 2A–D depicts four emergent few-shot prompted tasks from BIG-Bench, a crowd-sourced suite of over 200 benchmarks for language model evaluation (BIG-Bench, 2022). Figure 2A shows an arithmetic benchmark that tests 3-digit addition and subtraction, as well as 2-digit multiplication. GPT-3 and LaMDA (Thoppilan et al., 2022) have close-to-zero performance for several orders of magnitude of training compute, before performance jumps to sharply above random at $2 \cdot 10^{22}$ training FLOPs (13B parameters) for GPT-3, and $10^{23}$ training FLOPs (68B parameters) for LaMDA. Similar emergent behavior also occurs at around the same model scale for other tasks, such as transliterating from the International Phonetic Alphabet (Figure 2B), recovering a word from its scrambled letters (Figure 2C), and Persian question-answering (Figure 2D). Even more emergent abilities from BIG-Bench are given in Appendix E.

**TruthfulQA.** Figure 2E shows few-shot prompted performance on the TruthfulQA benchmark, which measures the ability to answer questions truthfully (Lin et al., 2021). This benchmark is adversarially curated against GPT-3 models, which do not perform above random, even when scaled to the largest model size. Small Gopher models also do not perform above random until scaled up to the largest model of $5 \cdot 10^{23}$ training FLOPs (280B parameters), for which performance jumps to more than 20% above random (Rae et al., 2021).

**Grounded conceptual mappings.** Figure 2F shows the task of grounded conceptual mappings, where language models must learn to map a conceptual domain, such as a cardinal direction, represented in a textual grid world (Patel & Pavlick, 2022). Again, performance only jumps to above random using the largest GPT-3 model.

**Multi-task language understanding.** Figure 2G shows the Massive Multi-task Language Understanding (MMLU) benchmark, which aggregates 57 tests covering a range of topics including math, history, law, and more (Hendrycks et al., 2021a). For GPT-3, Gopher, and Chinchilla, models of $\sim 10^{22}$ training FLOPs ($\sim$10B parameters) or smaller do not perform better than guessing on average over all the topics, scaling up to 3–5 $\cdot 10^{23}$ training FLOPs (70B–280B parameters) enables performance to substantially surpass random. This result is striking because it could imply that the ability to solve knowledge-based questions spanning a large collection of topics might require scaling up past this threshold (for dense language models without retrieval or access to external memory).

---

[2]Though GPT-3 popularized prompting, the task setup has existed since before GPT-3 (Trinh & Le, 2018; McCann et al., 2018; Radford et al., 2019; Raffel et al., 2020).

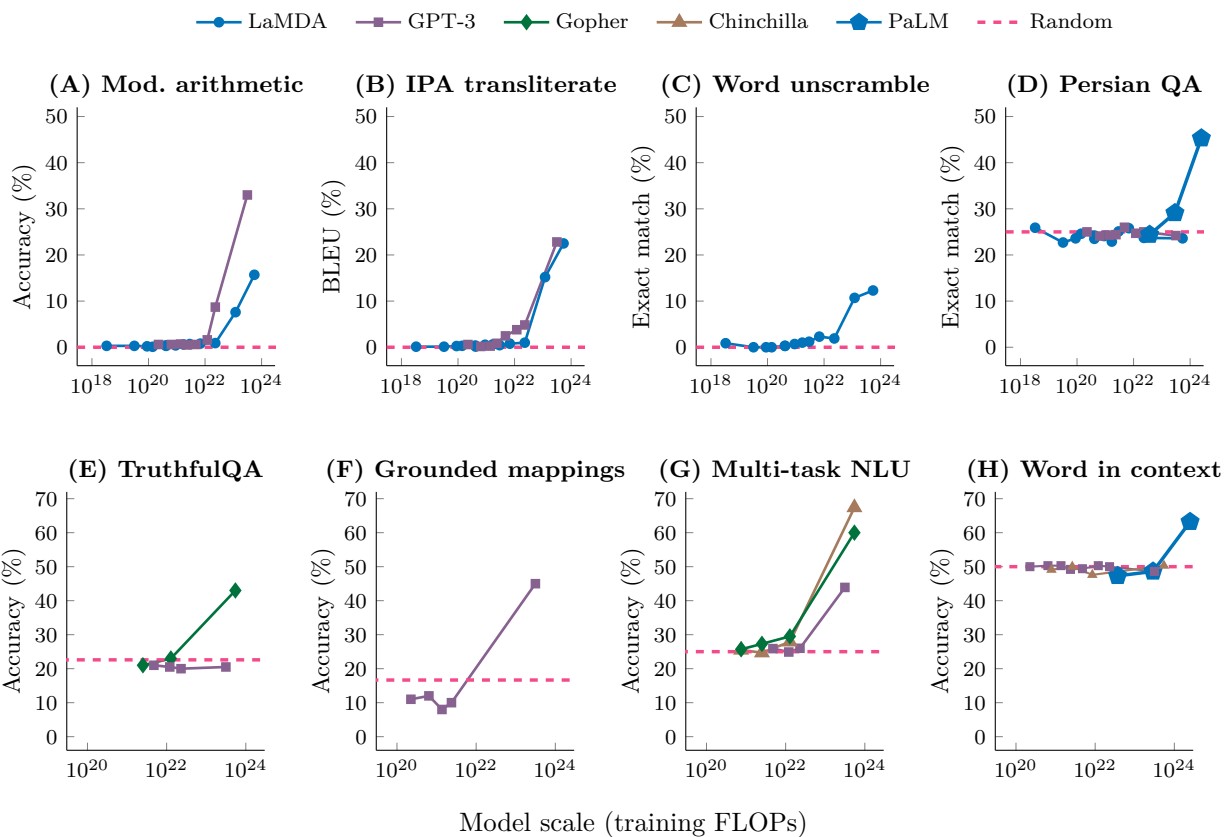

Figure 2: Eight examples of emergence in the few-shot prompting setting. Each point is a separate model. The ability to perform a task via few-shot prompting is emergent when a language model achieves random performance until a certain scale, after which performance significantly increases to well-above random. Note that models that used more training compute also typically have more parameters—hence, we show an analogous figure with number of model parameters instead of training FLOPs as the $x$-axis in Figure 11. A–D: BIG-Bench (2022), 2-shot. E: Lin et al. (2021) and Rae et al. (2021). F: Patel & Pavlick (2022). G: Hendrycks et al. (2021a), Rae et al. (2021), and Hoffmann et al. (2022). H: Brown et al. (2020), Hoffmann et al. (2022), and Chowdhery et al. (2022) on the WiC benchmark (Pilehvar & Camacho-Collados, 2019).

**Word in Context.** Finally, Figure 2H shows the Word in Context (WiC) benchmark (Pilehvar & Camacho-Collados, 2019), which is a semantic understanding benchmark. Notably, GPT-3 and Chinchilla fail to achieve one-shot performance of better than random, even when scaled to their largest model size of $\sim 5 \cdot 10^{23}$ FLOPs. Although these results so far may suggest that scaling alone may not enable models to solve WiC, above-random performance eventually emerged when PaLM was scaled to $2.5 \cdot 10^{24}$ FLOPs (540B parameters), which was much larger than GPT-3 and Chinchilla.

## 4 Augmented Prompting Strategies

Although few-shot prompting is perhaps currently the most common way of interacting with large language models, recent work has proposed several other prompting and finetuning strategies to further augment the abilities of language models. If a technique shows no improvement or is harmful when compared to the baseline of not using the technique until applied to a model of a large-enough scale, we also consider the technique an emergent ability.

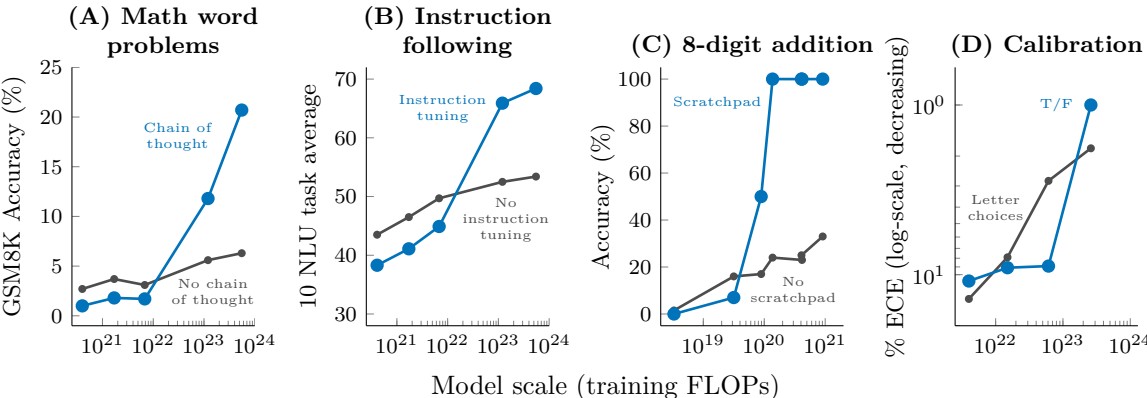

Figure 3: Specialized prompting or finetuning methods can be emergent in that they do not have a positive effect until a certain model scale. A: Wei et al. (2022b). B: Wei et al. (2022a). C: Nye et al. (2021). D: Kadavath et al. (2022). An analogous figure with number of parameters on the $x$-axis instead of training FLOPs is given in Figure 12. The model shown in A-C is LaMDA (Thoppilan et al., 2022), and the model shown in D is from Anthropic.

**Multi-step reasoning.** Reasoning tasks, especially those involving multiple steps, have been challenging for language models and NLP models more broadly (Rae et al., 2021; Bommasani et al., 2021; Nye et al., 2021). A recent prompting strategy called chain-of-thought prompting enables language models to solve such problems by guiding them to produce a sequence of intermediate steps before giving the final answer (Cobbe et al., 2021; Wei et al., 2022b; Zhou et al., 2022). As shown in Figure 3A, chain of thought prompting only surpasses standard prompting without intermediate steps when scaled to $10^{23}$ training FLOPs ($\sim$100B parameters). A similar emergence in performance gain was also observed when augmenting few-shot prompting with explanations that came after the final answer (Lampinen et al., 2022).

**Instruction following.** Another growing line of work aims to better enable language models to perform new tasks simply by reading instructions describing the task (without few-shot exemplars). By finetuning on a mixture of tasks phrased as instructions, language models have been shown to respond appropriately to instructions describing an unseen task (Ouyang et al., 2022; Wei et al., 2022a; Sanh et al., 2022). As shown in Figure 3B, Wei et al. (2022a) found that this instruction-finetuning technique hurts performance for models of $7 \cdot 10^{21}$ training FLOPs (8B parameters) or smaller, and only improves performance when scaled to $10^{23}$ training FLOPs ($\sim$100B parameters) (though Sanh et al. (2022) found shortly after that this instruction-following behavior could be also induced by finetuning smaller encoder-decoder T5 models).

**Program execution.** Consider computational tasks involving multiple steps, such as adding large numbers or executing computer programs. Nye et al. (2021) show that finetuning language models to predict intermediate outputs ("scratchpad") enables them to successfully execute such multi-step computations. As shown in Figure 3C, on 8-digit addition, using a scratchpad only helps for models of $\sim$9 $\cdot 10^{19}$ training FLOPs (40M parameters) or larger.

**Model calibration.** Finally, an important direction for deployment of language models studies is *calibration*, which measures whether models can predict which questions they will be able to answer correctly. Kadavath et al. (2022) compared two ways of measuring calibration: a True/False technique, where models first propose answers and then evaluate the probability "P(True)" that their answers are correct, and more-standard methods of calibration, which use the probability of the correct answer compared with other answer options. As shown in Figure 3D, the superiority of the True/False technique only emerges when scaled to the largest model scale of $\sim$3 $\cdot 10^{23}$ training FLOPs (52B parameters).

Table 1: List of emergent abilities of large language models and the scale (both training FLOPs and number of model parameters) at which the abilities emerge.

| | Emergent scale | | | |
| | Train. FLOPs | Params. | Model | Reference |
|---|---|---|---|---|
| **Few-shot prompting abilities** | | | | |
| • Addition/subtraction (3 digit) | 2.3E+22 | 13B | GPT-3 | Brown et al. (2020) |
| • Addition/subtraction (4-5 digit) | 3.1E+23 | 175B | | |
| • MMLU Benchmark (57 topic avg.) | 3.1E+23 | 175B | GPT-3 | Hendrycks et al. (2021a) |
| • Toxicity classification (CivilComments) | 1.3E+22 | 7.1B | Gopher | Rae et al. (2021) |
| • Truthfulness (Truthful QA) | 5.0E+23 | 280B | | |
| • MMLU Benchmark (26 topics) | 5.0E+23 | 280B | | |
| • Grounded conceptual mappings | 3.1E+23 | 175B | GPT-3 | Patel & Pavlick (2022) |
| • MMLU Benchmark (30 topics) | 5.0E+23 | 70B | Chinchilla | Hoffmann et al. (2022) |
| • Word in Context (WiC) benchmark | 2.5E+24 | 540B | PaLM | Chowdhery et al. (2022) |
| • Many BIG-Bench tasks (see Appendix E) | Many | Many | Many | BIG-Bench (2022) |
| **Augmented prompting abilities** | | | | |
| • Instruction following (finetuning) | 1.3E+23 | 68B | FLAN | Wei et al. (2022a) |
| • Scratchpad: 8-digit addition (finetuning) | 8.9E+19 | 40M | LaMDA | Nye et al. (2021) |
| • Using open-book knowledge for fact checking | 1.3E+22 | 7.1B | Gopher | Rae et al. (2021) |
| • Chain of thought: Math word problems | 1.3E+23 | 68B | LaMDA | Wei et al. (2022b) |
| • Chain of thought: StrategyQA | 2.9E+23 | 62B | PaLM | Chowdhery et al. (2022) |
| • Differentiable search index | 3.3E+22 | 11B | T5 | Tay et al. (2022) |
| • Self-consistency decoding | 1.3E+23 | 68B | LaMDA | Wang et al. (2022b) |
| • Leveraging explanations in prompting | 5.0E+23 | 280B | Gopher | Lampinen et al. (2022) |
| • Least-to-most prompting | 3.1E+23 | 175B | GPT-3 | Zhou et al. (2022) |
| • Zero-shot chain of thought reasoning | 3.1E+23 | 175B | GPT-3 | Kojima et al. (2022) |
| • Calibration via P(True) | 2.6E+23 | 52B | Anthropic | Kadavath et al. (2022) |

## 5 Discussion

We have seen that a range of abilities—in the few-shot prompting setup or otherwise—have thus far only been observed when evaluated on a sufficiently large language model. Hence, their emergence cannot be predicted by simply extrapolating performance on smaller-scale models. Emergent few-shot prompted tasks are also unpredictable in the sense that these tasks are not explicitly included in pre-training, and we likely do not know the full scope of few-shot prompted tasks that language models can perform. This raises the question of whether further scaling could potentially endow even-larger language models with new emergent abilities. Tasks that language models cannot currently do are prime candidates for future emergence; for instance, there are dozens of tasks in BIG-Bench for which even the largest GPT-3 and PaLM models do not achieve above-random performance (see Appendix E.4).

The ability for scale to unpredictably enable new techniques is not just theoretical. Consider the Word in Context (WiC) benchmark (Pilehvar & Camacho-Collados, 2019) shown in Figure 2H, as a historical example. Here, scaling GPT-3 to around $3 \cdot 10^{23}$ training FLOPs (175B parameters) failed to unlock above-random one-shot prompting performance.[3] Regarding this negative result, Brown et al. (2020) cited the model architecture of GPT-3 or the use of an autoregressive language modeling objective (rather than using a denoising training objective) as potential reasons, and suggested training a model of comparable size with bidirectional architecture as a remedy. However, later work found that further scaling a decoder-only language model was actually enough to enable above-random performance on this task. As is shown in Figure 2H, scaling PaLM (Chowdhery et al., 2022) from $3 \cdot 10^{23}$ training FLOPs (62B parameters) to $3 \cdot 10^{24}$ training FLOPs (540B parameters) led to a significant jump in performance, without the significant architectural changes suggested by Brown et al. (2020).

---

[3]GPT-3 does achieve slightly above-random performance on the dev set with few-shot instead of one-shot prompting ($\sim$55%), but this above-random performance did not appear to be a result of scale and did not hold on the test set server.

## 5.1 Potential explanations of emergence

Although there are dozens of examples of emergent abilities, there are currently few compelling explanations for why such abilities emerge in the way they do. For certain tasks, there may be natural intuitions for why emergence requires a model larger than a particular threshold scale. For instance, if a multi-step reasoning task requires $l$ steps of sequential computation, this might require a model with a depth of at least $O(l)$ layers. It is also reasonable to assume that more parameters and more training enable better memorization that could be helpful for tasks requiring world knowledge.[4] As an example, good performance on closed-book question-answering may require a model with enough parameters to capture the compressed knowledge base itself (though language model-based compressors can have higher compression ratios than conventional compressors (Bellard, 2021)).

It is also important to consider the evaluation metrics used to measure emergent abilities (BIG-Bench, 2022). For instance, using exact string match as the evaluation metric for long-sequence targets may disguise compounding incremental improvements as emergence. Similar logic may apply for multi-step or arithmetic reasoning problems, where models are only scored on whether they get the final answer to a multi-step problem correct, without any credit given to partially correct solutions. However, the jump in final answer accuracy does not explain why the quality of intermediate steps suddenly emerges to above random, and using evaluation metrics that do not give partial credit are at best an incomplete explanation, because emergent abilities are still observed on many classification tasks (e.g., the tasks in Figure 2D–H).

As an alternative evaluation, we measure cross-entropy loss, which is used in scaling laws for pre-training, for the six emergent BIG-Bench tasks, as detailed in Appendix A. This analysis follows the same experimental setup from BIG-Bench (2022) and affirms their conclusions for the six emergent tasks we consider. Namely, cross-entropy loss improves even for small model scales where the downstream metrics (exact match, BLEU, and accuracy) are close to random and do not improve, which shows that improvements in the log-likelihood of the target sequence can be masked by such downstream metrics. However, this analysis does not explain why downstream metrics are emergent or enable us to predict the scale at which emergence occurs. Overall, more work is needed to tease apart what enables scale to unlock emergent abilities.

## 5.2 Beyond scaling

Although we may observe an emergent ability to occur at a certain scale, it is possible that the ability could be later achieved at a smaller scale—in other words, model scale is not the singular factor for unlocking an emergent ability. As the science of training large language models progresses, certain abilities may be unlocked for smaller models with new architectures, higher-quality data, or improved training procedures. For example, there are 14 BIG-Bench tasks[5] for which LaMDA 137B and GPT-3 175B models perform at near-random, but PaLM 62B in fact achieves above-random performance, despite having fewer model parameters and training FLOPs. While there is not an empirical study ablating every difference between PaLM 62B and prior models (the computational cost would be too high), potential reasons for the better performance of PaLM could include high-quality training data (e.g., more multilingual and code data than LaMDA) and architectural differences (e.g., split digit-encodings; see Section 2 in Chowdhery et al. (2022)).

Moreover, once an ability is discovered, further research may make the ability available for smaller scale models. Consider the nascent direction of enabling language models to follow natural language instructions describing a task (Wei et al., 2022a; Sanh et al., 2022; Ouyang et al., 2022, *inter alia*). Although Wei et al. (2022a) initially found that instruction-based finetuning only worked for 68B parameter or larger decoder-only models, Sanh et al. (2022) induced similar behavior in a 11B model with an encoder-decoder architecture, which typically has higher performance after finetuning than decoder-only architectures (Wang et al., 2022a). As another example, Ouyang et al. (2022) proposed a finetuning and reinforcement learning from human feedback approach for the InstructGPT models, which enabled a 1.3B model to outperform much larger models in human-rater evaluations on a broad set of use cases.

---

[4]Though note that encoding world knowledge in parameters is just one approach; there are others (e.g., Guu et al., 2020; Borgeaud et al., 2021).

[5]These tasks are enumerated in Appendix F.

There has also been work on improving the general few-shot prompting abilities of language models (Gao et al., 2021; Schick & Schütze, 2021, *inter alia*). Theoretical and interpretability research (Wei et al., 2021a; Saunshi et al., 2021) on why a language modeling objective facilitates certain downstream behavior could in turn have implications on how to enable emergence beyond simply scaling. For instance, certain features of pre-training data (e.g., long-range coherence, having many rare classes) have also been shown to correlate with emergent few-shot prompting and could potentially enable it in smaller models (Xie et al., 2022; Chan et al., 2022), and few-shot learning can require certain model architectures in some scenarios (Chan et al., 2022). Computational linguistics work has further shown how threshold frequencies of training data can activate emergent syntactic rule-learning when model parameters and training FLOPs are held constant (Wei et al., 2021b), which has even been shown to have striking "aha" moments similar to those in the psycholinguistics literature (Abend et al., 2017; Zhang et al., 2021). As we continue to train ever-larger language models, lowering the scale threshold for emergent abilities will become more important for making research on such abilities to available to the community more broadly (Bommasani et al., 2021; Ganguli et al., 2022; Liang et al., 2022).

Naturally, there are limitations to a program consisting only of increasing scale (training compute, model parameters, and dataset size). For instance, scaling may eventually be bottle-necked by hardware constraints, and some abilities may not have emerged at this point. Other abilities may never emerge—for instance, tasks that are far out of the distribution of even a very large training dataset might not ever achieve any significant performance. Finally, an ability could emerge and then plateau; in other words, there is no guarantee that scaling enables an ability to reach the desired level.

### 5.3   Another view of emergence

While scale (e.g., training FLOPs or model parameters) has been highly correlated with language model performance on many downstream metrics so far, scale need not be the only lens to view emergent abilities. For example, the emergence of task-specific abilities can be analyzed as a function of the language model's perplexity on a general text corpus such as WikiText103 (Merity et al., 2016). Figure 4 shows such a plot with WikiText103 perplexity of the language model on the $x$-axis and performance on the MMLU benchmark on the $y$-axis, side-by-side with plots of training FLOPs and model parameters on the $x$-axis.

Because WikiText103 perplexity and training FLOPs happen to be highly correlated for the models considered here (Gopher and Chinchilla), the plots of emergent abilities look similar for both. However, this correlation between WikiText103 perplexity and scale may not hold in the future as new techniques beyond vanilla dense Transformer models are developed (e.g., retrieval-augmented models may have strong WikiText103 perplexity with less training compute and fewer model parameters (Borgeaud et al., 2021)). Also note that using WikiText103 perplexity to compare across model families can be complicated due to factors such as differences in training data composition. Overall, emergent abilities should probably be viewed as a function of many correlated variables.

### 5.4   Emergent risks

Importantly, similar to how emergent abilities have been observed in the few-shot prompting setting without explicitly being included in pre-training, risks could also emerge (Bommasani et al., 2021; Steinhardt, 2021; Ganguli et al., 2022). For instance, societal risks of large language models such as truthfulness, bias, and toxicity are a growing area of research (Weidinger et al., 2021). Such risks are important considerations whether or not they can be precisely characterized as "emergent" based on the definition in §2, and, in some scenarios, do increase with model scale (see the Inverse Scaling Prize[6]). Since work on emergent abilities incentivizes scaling language models, it is important to be aware of risks that increase with model scale even if they are not emergent.

Here, we summarize several prior findings on the relationship between specific social risks and model scale. On WinoGender (Rudinger et al., 2017), which measures gender bias in occupations such as "nurse" or "electrician," scaling has improved performance so far (Du et al., 2021; Chowdhery et al., 2022), though

---

[6]https://github.com/inverse-scaling/prize

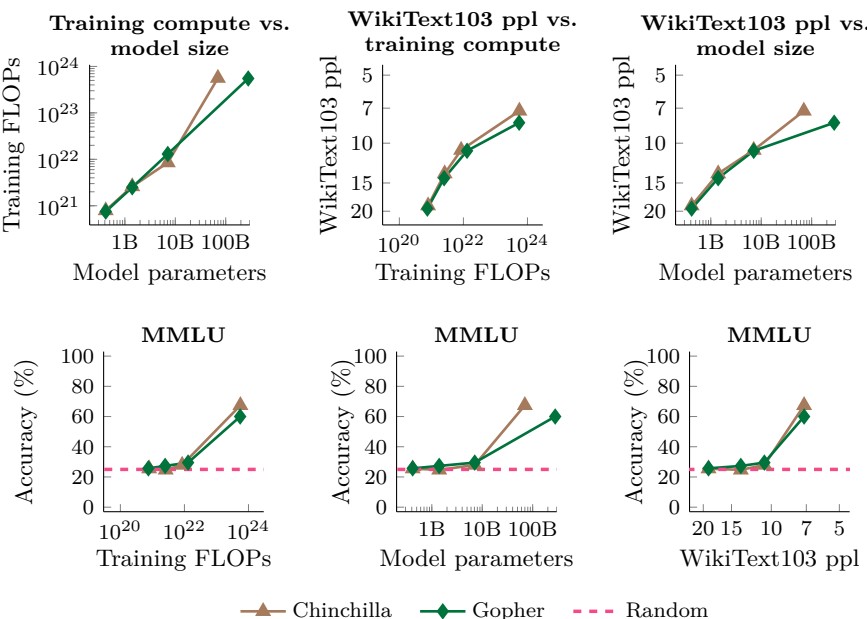

Figure 4: Top row: the relationships between training FLOPs, model parameters, and perplexity (ppl) on WikiText103 (Merity et al., 2016) for Chinchilla and Gopher. Bottom row: Overall performance on the massively multi-task language understanding benchmark (MMLU; Hendrycks et al., 2021a) as a function of training FLOPs, model parameters, and WikiText103 perplexity.

BIG-Bench (2022) found in BBQ bias benchmark (Parrish et al., 2022) that bias can increase with scaling for ambiguous contexts. As for toxicity, Askell et al. (2021) found that while larger language models could produce more toxic responses from the RealToxicityPrompts dataset (Gehman et al., 2020), this behavior could be mitigated by giving models prompts with examples of being "helpful, harmless, and honest." For extracting training data from language models, larger models were found to be more likely to memorize training data (Carlini et al., 2021; 2022), though deduplication methods have been proposed and can simultaneously reduce memorization while improving performance (Kandpal et al., 2022; Lee et al., 2022a). The TruthfulQA benchmark (Lin et al., 2021) showed that GPT-3 models were more likely to mimic human falsehoods as they got larger, though Rae et al. (2021) later showed on a multiple-choice version that scaling Gopher to 280B enabled emergent performance substantially better than random.

Beyond the above, emergent risks also include phenomena that might only exist in future language models or that have not yet been characterized in current language models. Some such behaviors, as discussed in detail in Hendrycks et al. (2021b), could be backdoor vulnerabilities, inadvertent deception, or harmful content synthesis. Approaches involving data filtering, forecasting, governance, and automatically discovering harmful behaviors have been proposed for discovering and mitigating emergent risks (Bender et al., 2021; Weidinger et al., 2021; Steinhardt, 2021; Ganguli et al., 2022; Perez et al., 2022, *inter alia*). For a more detailed discussion of the risks of large language models, including emergent risks, see Bender et al. (2021); Steinhardt (2021); Bommasani et al. (2021); Ganguli et al. (2022).

## 5.5 Sociological changes

Finally, the emergent abilities discussed here focus on model behavior and are just one of several types of emergence in NLP (Manning et al., 2020; Teehan et al., 2022). Another notable type of qualitative change is sociological, in which increasing scale has shifted how the community views and uses language models. For instance, NLP has historically focused on task-specific models (Jurafsky & Martin, 2009). Recently, scaling has led to an explosion in research on and development of models that are "general purpose" in that they are

single models that aim to perform a range of tasks not explicitly encoded in the training data (e.g., GPT-3, Chinchilla, and PaLM) (Manning, 2022).

One key set of results in the emergent sociological shift towards general-purpose models is when scaling enables a few-shot prompted general-purpose model to outperform prior state of the art held by finetuned task-specific models. As a few examples, GPT-3 175B achieved new state of the art on the TriviaQA and PiQA question-answering benchmarks (Brown et al., 2020); PaLM 540B achieved new state of the art on three arithmetic reasoning benchmarks (Chowdhery et al., 2022); and the multimodal Flamingo 80B model achieved new state of the art on six visual question answering benchmarks (Alayrac et al., 2022). In all of these cases, state-of-the-art performance was achieved by few-shot prompting a language model of unprecedented scale (scaling curves for these examples are shown in Appendix Figure 13). These abilities are not necessarily emergent since they have smooth, predictable scaling curves—however, they do underscore an emergent sociological shift towards general-purpose models in the NLP community.

The ability for general-purpose models to perform unseen tasks given only a few examples has also led to many new applications of language models outside the NLP research community. For instance, language models have been used via prompting to translate natural language instructions into actions executable by robots (Ahn et al., 2022; Huang et al., 2022), interact with users (Coenen et al., 2021; Wu et al., 2021; 2022a; Lee et al., 2022b), and facilitate multi-modal reasoning (Zeng et al., 2022; Alayrac et al., 2022). Large language models have also been deployed in the real-world both in products, such as GitHub CoPilot,[7] and directly as services themselves, such as OpenAI's GPT-3 API.[8]

### 5.6 Directions for future work

Future work on emergent abilities could involve train more-capable language models, as well as methods for better enabling language models to perform tasks. Some potential directions include but are not limited to the following.

**Further model scaling.** Further scaling up models has so far appeared to increase the capabilities of language models, and is a straightforward direction for future work. However, simply scaling up language models is computationally expensive and requires solving substantial hardware challenges, and so other approaches will likely play a key role in the future of the emergent abilities of large language models.

**Improved model architectures and training.** Improving model architecture and training procedures may facilitate high-quality models with emergent abilities while mitigating computational cost. One direction is using sparse mixture-of-experts architectures (Lepikhin et al., 2021; Fedus et al., 2021; Artetxe et al., 2021; Zoph et al., 2022), which scale up the number of parameters in a model while maintaining constant computational costs for an input. Other directions for better computational efficiency could involve variable amounts of compute for different inputs (Graves, 2016; Dehghani et al., 2018), using more localized learning strategies than backpropagation through all weights in a neural network (Jaderberg et al., 2017), and augmenting models with external memory (Guu et al., 2020; Borgeaud et al., 2021; Wu et al., 2022b, *inter alia*). These nascent directions have already shown promise in many settings but have not yet seen widespread adoption, which will likely require further work.

**Data scaling.** Training long enough on a large-enough dataset has been shown to be key for the ability of language models to acquire syntactic, semantic, and other world knowledge (Zhang et al., 2021; Wei et al., 2021b; Razeghi et al., 2022). Recently, Hoffmann et al. (2022) argued that prior work (Kaplan et al., 2020) underestimated the amount of training data needed to train a compute-optimal model, underscoring the importance of training data. Collecting large datasets so that models can be trained for longer could allow a greater range of emergent abilities under a fixed model size constraint.

**Better techniques for and understanding of prompting.** Although few-shot prompting (Brown et al., 2020) is simple and effective, general improvements to prompting may further expand the abilities of language models. For instance, simple modifications such as calibrating output probabilities (Zhao et al., 2021;

---

[7]https://copilot.github.com/
[8]https://beta.openai.com/docs/introduction

Holtzman et al., 2021) or using a noisy channel (Min et al., 2021) have improved performance on a range of tasks. Augmenting few-shot exemplars with intermediate steps (Reynolds & McDonell, 2021; Nye et al., 2021; Wei et al., 2022b) has also enabled models to perform multi-step reasoning tasks not possible in the standard prompting formulation from Brown et al. (2020). Moreover, better exploration of what makes prompting successful (Wei et al., 2021a; Xie et al., 2022; Min et al., 2022; Olsson et al., 2022) could lead to insights on how to elicit emergent abilities at a smaller model scale. Sufficient understanding of why models work generally lags the development and popularization of techniques such as few-shot prompting, and it is also likely that the best practices for prompting will change as more-powerful models are developed over time.

**Frontier tasks.** Although language models can perform a wide range of tasks, there are still many tasks that even the largest language models to date cannot perform with above-random accuracy. Dozens of such tasks from BIG-Bench are enumerated in Appendix E.4; these tasks often involve abstract reasoning (e.g., playing Chess, challenging math, etc). Future research could potentially investigate why these abilities have not yet emerged, and how to enable models to perform these tasks. Looking forward, another growing direction could be multilingual emergence; results on multilingual BIG-Bench tasks indicate that both model scale and training data play a role in emergence (e.g., Figure 2D shows that both using PaLM's training dataset and scaling to 62B parameters is required for question-answering in Persian). Other frontier tasks could include prompting in multiple modalities (Alayrac et al., 2022; Ramesh et al., 2022).

**Understanding emergence.** Beyond research on unlocking further emergent tasks, an important open question for future research is how and why emergent abilities occur in large language models. In this paper, we conducted initial analyses regarding scaling of the cross-entropy loss on BIG-Bench (Appendix A.1), different metrics for generative tasks (Appendix A.2), and for which types of tasks emergence occurs (Appendix A.3 and Appendix B). These analyses did not provide complete answers to why emergence occurs or how to predict it. Future research could potentially analyze emergence in new ways (e.g., analyze the relationship between emergent tasks and similar data in training; create a synthetic task that requires multiple compositional sub-tasks and evaluate how each of those sub-tasks improve with scale and unlock emergence when combined). Overall, understanding emergence is an important direction because it could potentially allow us predict what abilities future models may have, as well as provide new insights into how to train more-capable language models.

## 6    Conclusions

We have discussed emergent abilities of language models, for which meaningful performance has only been thus far observed at a certain computational scale. Emergent abilities can span a variety of language models, task types, and experimental scenarios. Such abilities are a recently discovered outcome of scaling up language models, and the questions of how they emerge and whether more scaling will enable further emergent abilities seem to be important future research directions for the field of NLP.

### Broader Impact Statement

In this paper, we surveyed results in the existing literature, without proposing new methods or models. As discussed in (§5), emergent abilities are unpredictable in several ways, and include emergent risks (§5.4). We believe these phenomena warrant careful study and raise important questions for the field.

### Acknowledgments

We thank Charles Sutton, Slav Petrov, Douglas Eck, Jason Freidenfelds, Jascha Sohl-Dickstein, Ethan Dyer, Dale Schuurmans, and Xavier Garcia for useful discussions and feedback on the manuscript.

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

# A    BIG-Bench analysis

## A.1    Cross-entropy loss analysis

Here we study how scaling curves may appear differently depending on the evaluation metric used to measure performance. We will focus on the six few-shot prompted BIG-Bench tasks that we consider emergent for LaMDA models. Three of these tasks are generative and use Exact Match (EM) or BLEU (Papineni et al., 2002) as the evaluation metric. The other three tasks are classification and use accuracy (acc) as the evaluation metric.

In the scaling curves for these tasks, peformance in EM/BLEU/acc is close to random for small models ($\leq 10^{22}$ FLOPs / $\leq$27B params). We will compare these scaling curves against alternative plots that have a different $y$-axis measured by cross-entropy loss. Cross-entropy loss differs from EM/BLEU/acc in that it captures improvements in performance (the predicted distribution getting closer to ground truth) even when the EM/BLEU/acc is random. For example, if two examples are both wrong as measured by EM/BLEU/acc, one example may be closer to the ground truth in terms of probabilities, and this information is captured by the cross-entropy loss.

These plots are expected to look like one of the following:

- Outcome 1: For the model scales where EM/BLEU/acc is random, cross-entropy loss also does not improve as scale increases. This outcome implies that for these scales, the model truly does not get any better at the tasks.

- Outcome 2: For the model scales where EM/BLEU/acc is random, cross-entropy loss does improve. This outcome implies that the models do get better at the task, but these improvements are not reflected in the downstream metric of interest. The broader implication is that scaling small models improves the models in a way that is not reflected in EM/BLEU/Acc, and that there is some critical model scale where these improvements enable the downstream metric to increase to above random as an emergent ability.

We find that all six BIG-Bench tasks fall under Outcome 2, and detail this analysis below. Overall, the conclusion from this analysis is that small models do improve in some ways that downstream metrics that EM/BLEU/Acc do not capture. However, these tasks are still considered emergent, and this analysis does not provide any straightforward indicators of how to predict such emergent behaviors.

### A.1.1    Generative tasks

Figure 5 shows the cross-entropy loss on the three generative BIG-Bench tasks (modified arithmetic, IPA transliterate, and word unscramble) alongside the downstream evaluation metrics used in Figure 2. For all three tasks, notice that while the error rate is nearly 100% for small models ($\leq 10^{22}$ FLOPs / $\leq$27B params), the cross-entropy loss does actually improve for these model sizes. At the point of emergence as measured by error rate, we also see an "elbow" in performance improvement for cross-entropy loss.

### A.1.2    Classification tasks

Figure 6 (middle row) shows the cross-entropy loss of the three classification BIG-Bench tasks. Similar to the generative tasks, when the error rate is close to random, cross-entropy loss consistently still improves for models trained with more compute. This again shows that performance as computed by accuracy can mask consistent improvements in the likelihood of the target sequences.

We also perform an additional analysis of the multiple choice emergent tasks in Figure 6 (bottom row), which shows the log probabilities of the correct response and incorrect response(s). We find that the cross-entropy loss decreases for both the correct and incorrect responses in the three emergent multiple choice tasks. Counterintuitively, both log-probabilities can decrease in tandem even when the probability across all available multiple choice responses is normalized. The reason is that larger models produce less-extreme probabilities (i.e., values approaching 0 or 1) and therefore the average log-probabilities have fewer extremely

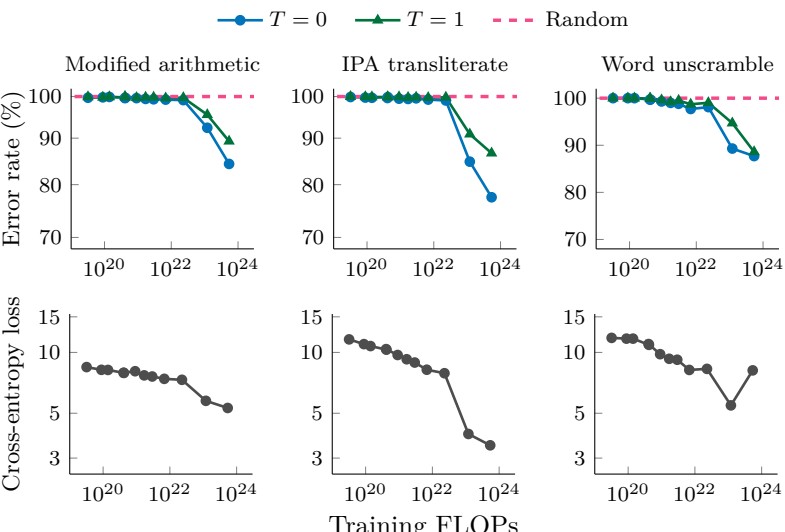

Figure 5: Adjacent plots for error rate and cross-entropy loss on three emergent generative tasks in BIG-Bench for LaMDA. We show error rate for both greedy decoding ($T = 0$) as well as random sampling ($T = 1$). Error rate is (1 - exact match score) for modified arithmetic and word unscramble, and (1 - BLEU score) for IPA transliterate.

small values. However, we note that for each of these three tasks, that the average log-probability of the correct and incorrect responses eventually deviates at a certain scale, during which performance on the task increases substantially.

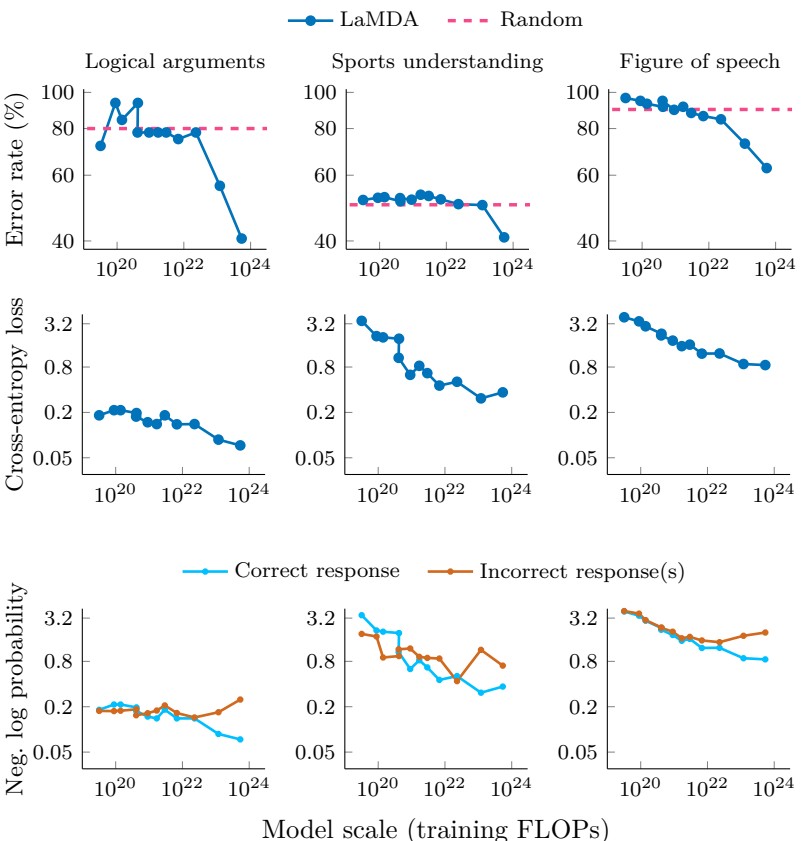

Figure 6: Adjacent plots for error rate, cross-entropy loss, and log probabilities of correct and incorrect responses on three classification tasks on BIG-Bench that we consider to demonstrate emergent abilities. Logical arguments only has 32 samples, which may contribute to noise. Error rate is (1 - accuracy).

## A.2 Different metrics for generative tasks

In §5.1 we asked whether the apparent emergent abilities on generative tasks were due to using a particular metric such as exact string match, which does not award partially correct sequences. Here, we show three emergent generative BIG-Bench tasks using all evaluation metrics provided by BIG-Bench, which includes metrics such as BLEU, ROUGE, and BLEURT, that award partial credit for answers that do not exactly match the target. For all three tasks, the emergent behavior appears to be independent of which evaluation metric is used. Hence, we conclude that using exact string match instead of another evaluation metric that awards partial credit is not a complete explanation of emergence on generative tasks. Two emergent generative BIG-Bench tasks, word unscramble and repeat copy logic, are excluded here because exact match is the only most sensible evaluation metric for those tasks, which measure the ability to manipulate words in the input (and hence metrics like BLEU and ROUGE that give word-level partial credit are not valid).

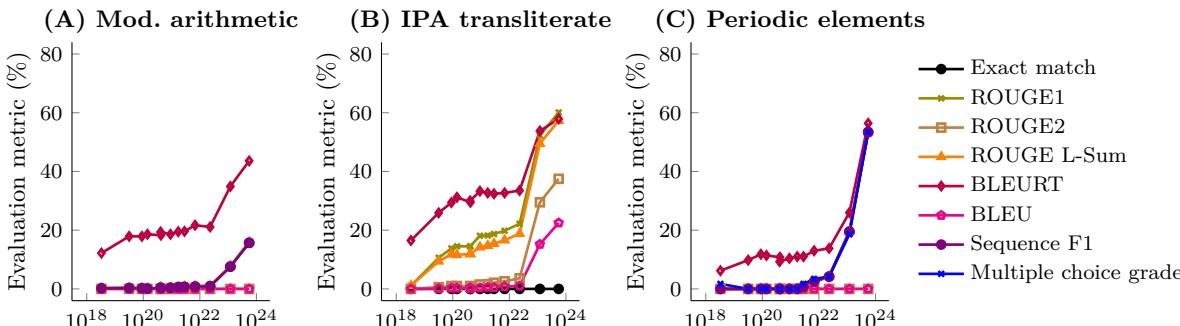

Figure 7: Multiple evaluation metrics for emergent BIG-Bench tasks that are generative in nature. For all three tasks, emergent behavior is apparent for all evaluation metrics.

## A.3 BIG-Bench task analysis

BIG-Bench contains over 200 tasks, and each task has associated keywords identified by the authors who submitted the task (e.g., "common sense", "multilingual"). Given this, we asked the question, which types of BIG-Bench tasks are more likely to be emergent (compared with scaling smoothly)? For this analysis, we manually classified all 210 BIG-Bench tasks as thus far emergent or not. We used the definition of emergence given in §3, which is that the task had near-random performance until a certain scale, after which performance increases to substantially above random (as opposed to smoothly increasing). Because this definition is potentially subjective based on the definition of "near-random" (and any heuristic we decide on would encode these subjective biases), two co-authors of the paper worked together and agreed with confidence on all the tasks labeled as emergent. For full transparency, this set of annotations is listed in Appendix E.

In Figure 8, we show the number of tasks that are emergent for each keyword in BIG-Bench. Furthermore, we stratify them by tasks that first emerged with LaMDA 137B or GPT-3 175B, as well as tasks that were not emergent until using PaLM models. The non-emergent tasks in this plot include either "smoothly increasing" tasks (performance predictably increased with model size) or "flat" tasks (all models achieved approximately random performance). The remaining 40 BIG-Bench tasks not included in this chart did not fit into any of the above categories (e.g., too noisy due to very few eval examples, performance not correlated with model scale, etc.).

Since the number of tasks per keyword varied substantially among keywords, and most keywords had less than twenty tasks, the "most emergent" keywords differed depending on whether we compare number of emergent tasks or percentage of emergent tasks per keyword. Tracking the absolute number of emergent tasks per keyword is problematic since it effectively just captures the most common keywords used across BigBench. We therefore tracked which keywords had the highest percent of emergent tasks, which were analogical reasoning, word sense disambiguation, truthfulness, social reasoning, and emotional understanding. While one might expect a priori that reasoning-related tasks would more likely to be emergent, only two of the top five tasks were reasoning and other keyword tags like logical reasoning and causal reasoning did not

have a particularly high fraction of emergent tasks. Moreover, arithmetic and mathematics had relatively low percentage of emergent tasks, which was unexpected since some of the earliest examples of emergence were on arithmetic (Brown et al., 2020). Overall, there are no clear trends for which types of tasks are most emergent.

Finally, examining which keywords have the most tasks with flat scaling curves can also align with prior intuitions. For instance, visual reasoning has the largest fraction of tasks with flat scaling curves (8/13), since language models are not designed for visual reasoning. Other categories with a large fraction of flat scaling curve tasks are non-language, repeated interaction, context length, computer code, and multi-step—all targeting weaknesses of large language models. These flat categories could be directions for future work in emergence in large language models.

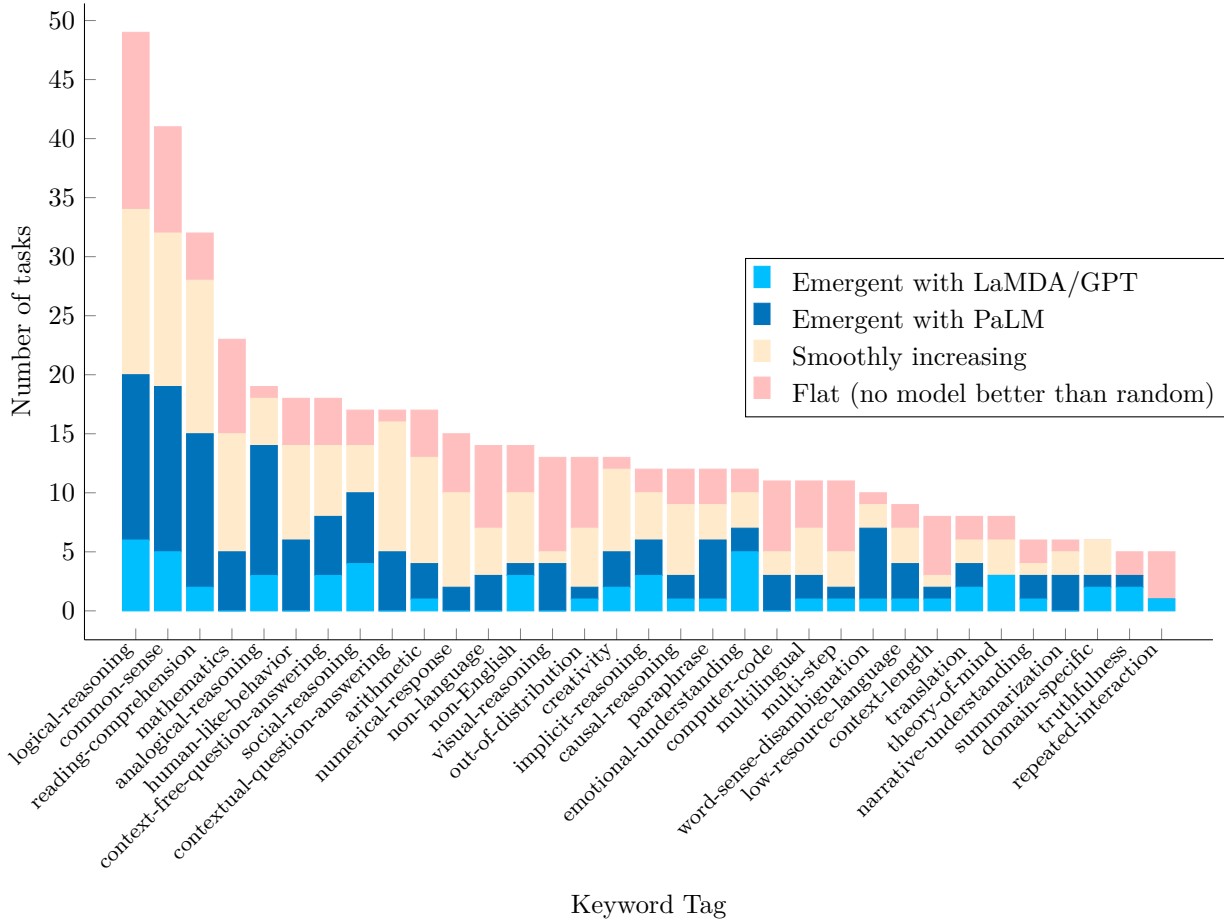

Figure 8: Proportion of emergent tasks for keywords in BIG-Bench (each task can be associated with multiple keywords). We only included keywords with at least five tasks. Smoothly increasing: performance improved predictably as model scale increased. Emergent with LaMDA/GPT: performance was near-random until used with LaMDA 137B or GPT-3 175B. Emergent with PaLM: performance was near-random for all previous models, until using a PaLM model (8B, 62B, or 540B). Flat: no model performs better than random.

# B   Further MMLU analysis

In §5.3, we saw how emergent performance on MMLU for Gopher and Chinchilla could be viewed as a function of training FLOPs, model parameters, and WikiText103 perplexity. Because MMLU is actually a suite of 57 topics spanning four categories, we ask the question of whether certain categories were more conducive to emergence than others. This is similar in nature to the BIG-Bench analysis done in the prior section (Appendix A.3). One difference here is that the MMLU categories are mutually exclusive—each topic only has one category, whereas a single BIG-Bench task often had multiple keyword tags. However, there are only four categories and 57 tasks for MMLU (compared with 200+ tasks and dozens of keywords for BIG-Bench).

In Figure 10, we stratify the performance of MMLU among the four categories given in the benchmark (Humanities, STEM, Social Science, and other), and plot them with multiple $x$-axes: training FLOPs, model parameters, and WikiText103 perplexity. It is clear that Social Science and Humanities have the largest jump in performance from the second-largest to the largest model, and STEM has the smallest jump in performance. For a given $x$-axis (training FLOPs, model parameters, WikiText103 ppl), all four categories had similar plot shapes. This result is also summarized in Figure 9.

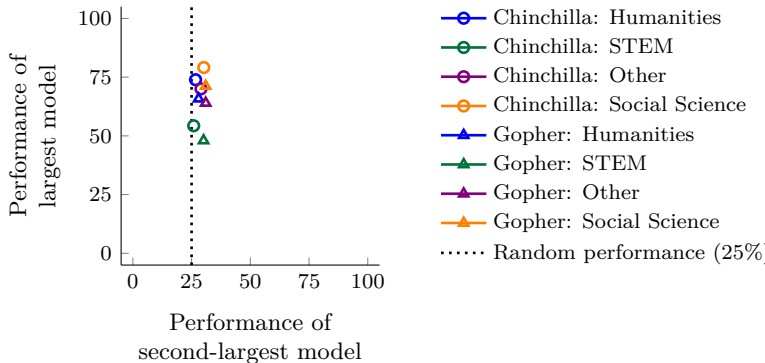

Figure 9: Performance of largest Chinchilla and Gopher models (70B and 280B, respectively) compared with the second-largest model (7B parameters for both Chiinchlla and Gopher). The 7B Chinchilla and Gopher models perform around random (25%) for all four categories. So the categories that improved the most from 7B to 70B/280B are humanities and social science, whereas STEM (Science, Technology, Engineering, and Mathematics) improved the least.

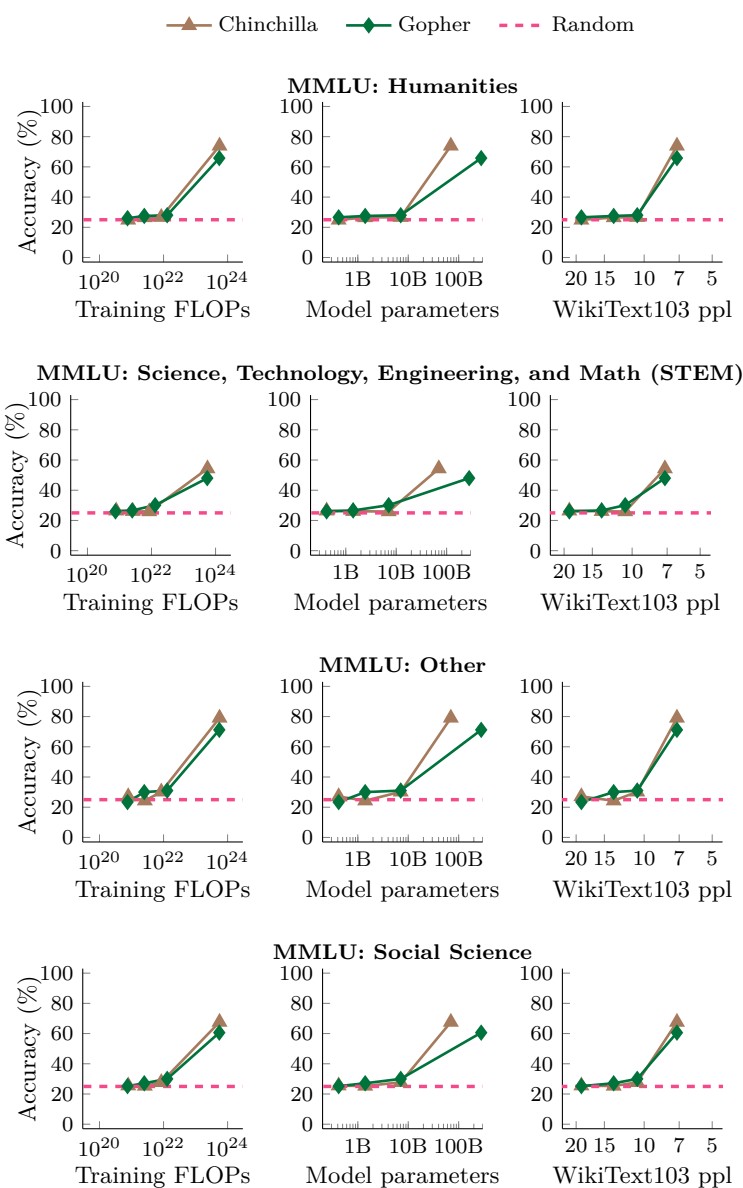

Figure 10: Emergence of Chinchilla and Gopher on MMLU. In the four rows, performance is stratified into four supercategories. For both Chinchilla and Gopher, Social Science had the highest level of emergence while STEM was the least emergent.

## C  All Model Details

Table 2 below summarizes the parameter count, number of training tokens, and the training FLOPs for the models highlighted in our work. The models span from the smallest LaMDA model with 2.1M parameters to the largest PaLM model with 540B parameters and 2.5E+24 training FLOPs—roughly 8x the computational budget of GPT-3.

Table 2: Parameters, training examples, and training FLOPs of large language models.

| Model | Parameters | Train tokens | Train FLOPs |
|---|---|---|---|
| GPT-3 | 125M | 300B | 2.25E+20 |
|  | 350M | 300B | 6.41E+20 |
|  | 760M | 300B | 1.37E+21 |
|  | 1.3B | 300B | 2.38E+21 |
|  | 2.7B | 300B | 4.77E+21 |
|  | 6.7B | 300B | 1.20E+22 |
|  | 13B | 300B | 2.31E+22 |
|  | 175B | 300B | 3.14E+23 |
| LaMDA | 2.1M | 262B | 3.30E+18 |
|  | 17M | 313B | 3.16E+19 |
|  | 57M | 262B | 8.90E+19 |
|  | 134M | 170B | 1.37E+20 |
|  | 262M | 264B | 4.16E+20 |
|  | 453M | 150B | 4.08E+20 |
|  | 1.1B | 142B | 9.11E+20 |
|  | 2.1B | 137B | 1.72E+21 |
|  | 3.6B | 136B | 2.96E+21 |
|  | 8.6B | 132B | 6.78E+21 |
|  | 29B | 132B | 2.30E+22 |
|  | 69B | 292B | 1.20E+23 |
|  | 137B | 674B | 5.54E+23 |
| Gopher | 417M | 300B | 7.51E+20 |
|  | 1.4B | 300B | 2.52E+21 |
|  | 7.1B | 300B | 1.28E+22 |
|  | 280B | 325B | 5.46E+23 |
| Chinchilla | 417M | 314B | 7.86E+20 |
|  | 1.4B | 314B | 2.63E+21 |
|  | 7.1B | [sic] 199B | 8.47E+21 |
|  | 70B | 1.34T | 5.63E+23 |
| PaLM | 8B | 780B | 3.74E+22 |
|  | 62B | 780B | 2.90E+23 |
|  | 540B | 780B | 2.53E+24 |
| Anthropic LM | 800M | 850B | 4.08E+21 |
|  | 3B | 850B | 1.53E+22 |
|  | 12B | 850B | 6.12E+22 |
|  | 52B | 850B | 2.65E+22 |

# D    Scaling with Parameter Count

Figures 11, 12, and 13 shows emergent abilities with an $x$-axis of number of model parameters.

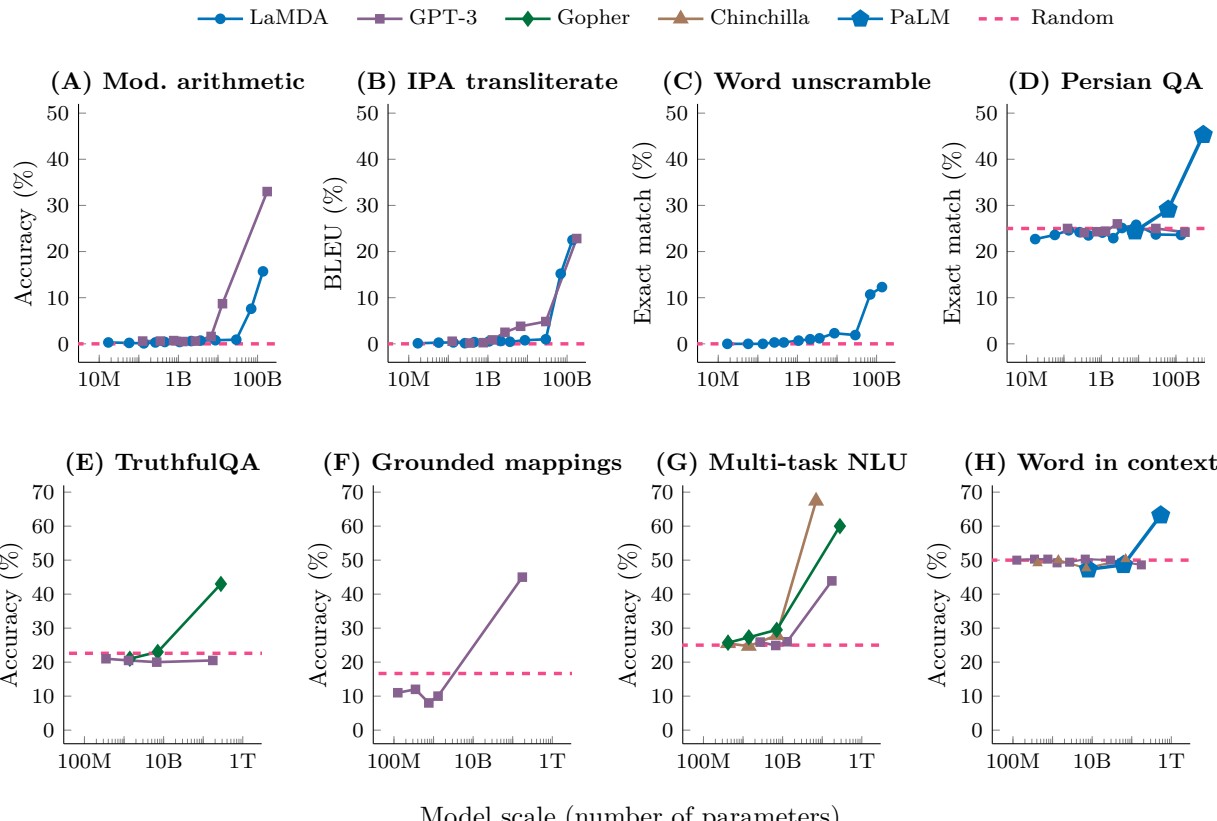

Figure 11: Eight examples of emergence in the few-shot prompting setting. Each point is a separate model. The ability to perform a task via few-shot prompting is emergent when a language model achieves random performance until a certain scale, after which performance significantly increases to well-above random. Note that models with more parameters also typically use more training compute—hence, we show an analogous figure with training FLOPs instead of number of model parameters as the $x$-axis in Figure 2. A–D: BIG-Bench (2022), 2-shot. E: Lin et al. (2021) and Rae et al. (2021). F: Patel & Pavlick (2022). G: Hendrycks et al. (2021a), Rae et al. (2021), and Hoffmann et al. (2022). H: Brown et al. (2020), Hoffmann et al. (2022), and Chowdhery et al. (2022) on the WiC benchmark (Pilehvar & Camacho-Collados, 2019).

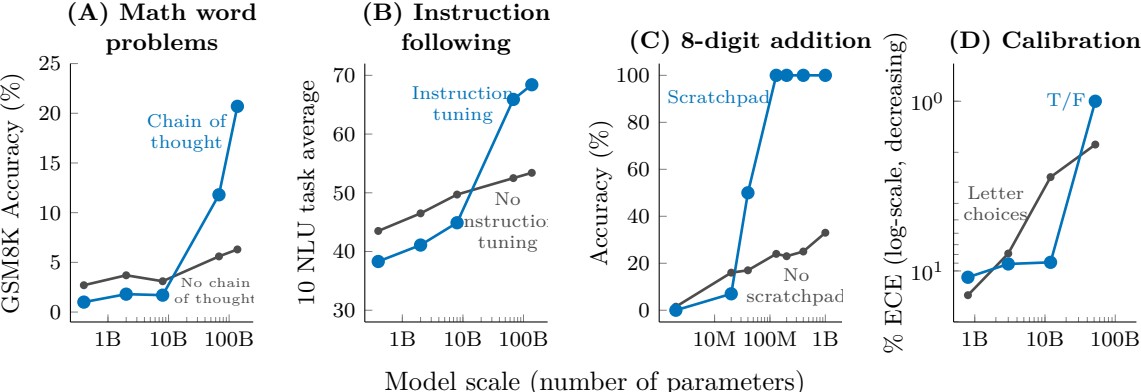

Figure 12: Specialized prompting or finetuning methods can be emergent in that they do not have a positive effect until a certain model scale. A: Wei et al. (2022b). B: Wei et al. (2022a). C: Nye et al. (2021). D: Kadavath et al. (2022). The model shown in A-C is LaMDA (Thoppilan et al., 2022), and the model shown in D is from Anthropic.

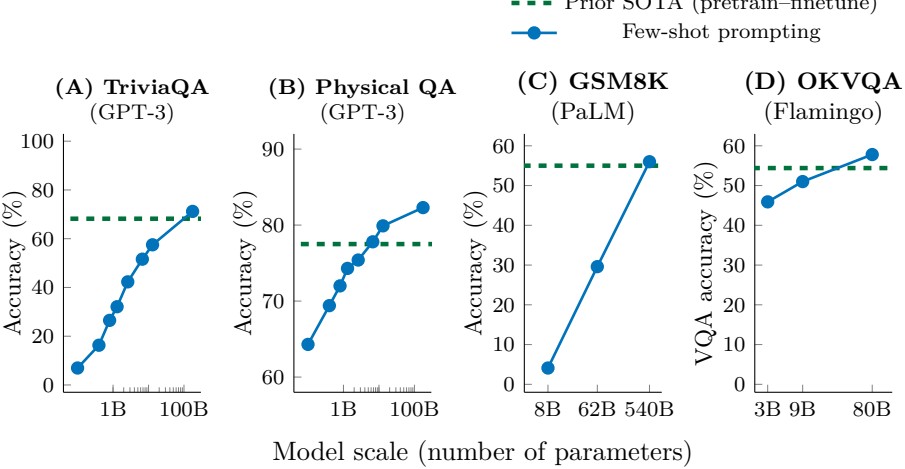

Figure 13: On some benchmarks, task-general models (not explicitly trained to perform a task) surpass prior state-of-the-art performance held by a task-specific model. A & B: Brown et al. (2020). C: Chowdhery et al. (2022). D: Alayrac et al. (2022).

# E  BIG-Bench Task Classification

This appendix contains the task classification annotations used for Figure 8 in Appendix A.3. Each task only appears in a single category. That is, if a task was initially emergent with GPT-3 or LaMDA, we excluded it from the PaLM emergence category.

Notably, Appendix E.4 lists the tasks where no model performs better than random (i.e., flat scaling curve). These tasks are potential candidates for future emergence, since a model in the future might achieve above-random performance on them.

## E.1  Smoothly increasing

abstract narrative understanding, auto categorization, bbq lite json, cause and effect, chess state tracking, conlang translation, context definition alignment, contextual parametric knowledge conflicts, coqa conversational question answering, cryobiology spanish, date understanding, emojis emotion prediction, empirical judgments, entailed polarity, evaluating information essentiality, forecasting subquestions, gem, general knowledge, hindi question answering, human organs senses, implicatures, implicit relations, intent recognition, linguistic mappings, list functions, matrixshapes, mult data wrangling, multiemo, natural instructions, nonsense words grammar, object counting, operators, penguins in a table, physics, polish sequence labeling, qa wikidata, reasoning about colored objects, rephrase, riddle sense, sentence ambiguity, similarities abstraction, simp turing concept, simple arithmetic, simple arithmetic json, simple arithmetic json multiple choice, simple arithmetic json subtasks, simple arithmetic multiple targets json, simple ethical questions, squad shifts, subject verb agreement, swedish to german proverbs, undo permutation, unit conversion, unnatural in context learning, bridging anaphora resolution barqa, disfl qa, novel concepts, periodic elements

## E.2  Emergent with GPT-3 or LaMDA

analytic entailment, codenames, common morpheme, fact checker, figure of speech detection, gender inclusive sentences german, hindu knowledge, international phonetic alphabet transliterate, irony identification, logical args, logical deduction, misconceptions, modified arithmetic, phrase relatedness, physical intuition, question answer creation, repeat copy logic, self evaluation tutoring, social iqa, sports understanding, strange stories, strategyqa, swahili english proverbs, word sorting, word unscrambling

## E.3  Emergent wih PaLM

anachronisms, analogical similarity, ascii word recognition, auto debugging, causal judgment, code line description, conceptual combinations, crass ai, cryptonite, cs algorithms, disambiguation qa, elementary math qa, emoji movie, english proverbs, english russian proverbs, geometric shapes, goal step wikihow, gre reading comprehension, hinglish toxicity, hyperbaton, identify odd metaphor, international phonetic alphabet nli, language identification, linguistics puzzles, logic grid puzzle, logical fallacy detection, logical sequence, metaphor boolean, metaphor understanding, movie dialog same or different, odd one out, parsinlu qa, parsinlu reading comprehension, physics questions, question selection, snarks, sufficient information, temporal sequences, timedial, understanding fables, unit interpretation, vitaminc fact verification

## E.4  Flat (no model better than random)

abstraction and reasoning corpus, authorship verification, checkmate in one, chinese remainder theorem, cifar10 classification, color, com2sense, cycled letters, discourse marker prediction, formal fallacies syllogisms negation, hhh alignment, kanji ascii, kannada, key value maps, language games, mathematical induction, minute mysteries qa, misconceptions russian, mnist ascii, multistep arithmetic, navigate, paragraph segmentation, play dialog same or different, presuppositions as nli, program synthesis, python programming challenge, real or fake text, roots optimization and games, salient translation error detection, self awareness, semantic parsing in context sparc, semantic parsing spider, simple text editing, sudoku, symbol interpretation, talkdown, tense, text navigation game, topical chat, tracking shuffled objects, twenty questions, web of lies, which wiki edit, winowhy, word problems on sets and graphs

### E.5 Other

Better than random and not correlated with scale: boolean expressions, crash blossom, dynamic counting, entailed polarity hindi, epistemic reasoning, factuality of summary, fantasy reasoning, gender sensitivity chinese, gender sensitivity english, high low game, identify math theorems, intersect geometry, muslim violence bias, persian idioms, protein interacting sites, scientific press release, self evaluation courtroom, social support, spelling bee, taboo, training on test set, truthful qa, yes no black white, dark humor detection, dyck languages, moral permissibility, ruin names

Model gets worse with scale: bbq lite, bias from probabilities, diverse social bias, movie recommendation, unqover

Not enough examples: known unknowns, suicide risk, what is the tao

Incomplete evals: convinceme, long context integration, medical questions russian

Other: arithmetic (emergent at 1B, which is none of the above categories), few-shot nlg (not sure why BLEURT is negative here)

## F  PaLM 62B is emergent but GPT-3 and LaMDA are not

We made the point in §5.2 that scale is not the only factor in emergence, since PaLM 62B shows emergence on many BIG-Bench tasks for which GPT-3 175B and LaMDA 137B do not, even though PaLM 62B has fewer model parameter and less training FLOPs.

This is the list of tasks: anachronisms, ascii word recognition, conceptual combinations, cryptonite, disambiguation qa, emoji movie, goal step wikihow, gre reading comprehension, linguistics puzzles, logic grid puzzle, metaphor boolean, metaphor understanding, odd one out, parsinlu qa.

