# OpenReview forum: "Emergent Abilities of Large Language Models"
_TMLR — Accepted by TMLR_

### Review · Reviewer_G7uJ · 2022-07-09

**Summary Of Contributions:**

This paper is a combination of a survey and an opinion piece, where the main ciam is: “language models exhibit emergent abilities—discontinuous improvements as scale increases, and this suggests additional scaling will unblock unpredictable capacity of language models”.

The first part of the paper surveys results from existing papers that show discontinuous improvements on a range of end tasks as scale increases, in language model prompting, chain-of-thought reasoning and instruction tuning.

The second part of the paper includes broader implications, including what would lead to this behavior, how this phenomenon could appear in smaller-scale models as well, and how this phenomenon has changed the application in NLP.


**Broader Impact Concerns:**

Section 5.3 briefly discusses emergent risks, focusing on security issues in extraction of training data. The paper should include (self-contained) discussion on broader risks in applying large language models, including models having bias/harmful contents.

**Requested Changes:**

* A broader overview on when emergent abilities occur / do not occur (e.g., in terms of tasks, models) that is beyond what has been already reported in prior work
* More systematic analysis on how exactly emergent abilities occur
* More systematic hypotheses & empirical evidence on why emergent abilities occur, i.e., what leads to discontinuous improvements as scale increases
* More systematic survey & empirical evidence on the impact of evaluation metrics in evaluating emergent abilities, beyond what has been already reported in prior work (could be elaboration from current Section 5.1/5.3)
* More systematic survey & empirical evidence on factors for emergent abilities beyond scale (could be elaborated from current Section 5.2)
* More fundamental evidence on the claim "further scaling will likely endow even-larger language models with new emergent abilities" with more concrete discussion (with possibly evidence) on how those new emergent abilities would look like and how further scaling will be possibly in a approachable way.

**Strengths And Weaknesses:**

**Strengths**
The paper includes a survey on discontiguous improvements made by large language models, aggregating results from multiple papers in a compact and coherent manner. The key message is very clear while presenting these results.
The paper discusses a range of broader implications including how this phenomenon can be applied to smaller-scale models, potential risk, sociological implication and more.

**Key weakness**
(As I detailed below) the paper does not have sufficient originality in the sense that it does not provide any new results/indications/insights on behaviors of language models. The paper largely focuses on showing how much emergence occurs in a “sudden” manner, bringing reports from previous work. It relies on the “magic” of emergence, rather than providing new insights on why this is happening and when it happens/does not happen.

In order to be accepted, I believe there should be more fundamental changes made, as I described in the "Requested Changes" section.

**Detailed weaknesses**
* The survey part of the paper is simply an aggregation of results reported in previous work without providing new axes of results or new insights. In most cases, exact results and indications are already reported and discussed in the cited papers. To point out a few:
Most figures in Figure 2 are exact copy of figures from the cited papers, e.g., the BigBench paper (Figure App.4)
    * Figure 3A is an exact copy of one of Figure 4 in [Wei et al. 2022. C], and the same discussion is made in this paper: “Notably, chain of thought reasoning is an emergent ability of increasing model scale.”
    * Figure 3B is an exact copy of Figure 7 in [Wei et al. 2022. F], and the same discussion is made in this paper: “the benefits of instruction tuning emerge only with sufficient model scale.”
    * Figure 3C and D is an (almost) exact copy of Figure 3.
* The survey part of the paper does not include new discussion or provide new insights on such discontinuous improvements, apart from simply reporting them that are already reported in the cited papers. For instance, the paper could have included results or analysis on different axes of scale (e.g. model parameters, pre-training data size, etc, apart from FLOP), or compare results in different metrics (in light of discussion in Section 5.1) to include more precise evidence for claims in Section 5.
* Most claims in Section 5 are not grounded by real evidence reported in the paper, only hypothetical, or too generic. To point out a few:
    * “The overall implication is that further scaling will likely endow even-larger language models with new emergent abilities.” – This is one of the main claims also mentioned in the abstract, and examples in Section 4 are provided as evidence. I do not believe this can be evidence, although it can be an example. The paper does not discuss potential new abilities that current largest models do not have but larger models in the future may potentially have. The paper does not discuss how the larger scale can be achieved, given that scaling the currently-dominated, dense language has been claimed to be non-trivial.
    * First paragraph of Section 5.1: I found these explanations are too loose to be a convincing explanation of emergent behaviors. An l-step reasoning task requiring at least O(l) layers is technically correct, but we already know 2 layers are not enough for two-step reasoning tasks. The aspect of knowledge memorization explains performance increasing linearly, but does not explain performance increasing discontinuously.
    * The rest of paragraphs in Section 5.1 are not about “Potential explanations of emergence”. These discussions are more about the limitations on evaluating models, which should be included in Section 4, potentially with how different metrics change the result reported in Section 4. (And this applies to Section 5.3.)
    * Section 5.2 is a pretty generic survey on other factors of language models aside from scale. It seems only loosely related, unless the connection to emergent abilities of language models is discussed or demonstrated more tightly.

**References**
Wei et al. 2022. F: https://arxiv.org/abs/2109.01652
Wei et al. 2022. C: https://arxiv.org/abs/2201.11903
Nye et al. 2021. https://openreview.net/forum?id=iedYJm92o0a
BigBench 2022: https://arxiv.org/abs/2206.04615

**Update after author responses**
Thank you for extensive responses and updates based on my review. I really appreciate authors' detailed responses, and I think most changes I requested (other than more open-ended requests) were made, e.g., adding more discussion, more empirical analysis with respect to different training budget, different metrics, etc. I must say I still have doubt whether this satisfies TMLR's criteria on survey paperson "drawing *new, previously unreported* connections between several pieces of work in an area" since I still believe the phenomenon of discontinuous improvements on various end tasks as models scale has been already reported in many papers. However, I agree that this paper did great job in aggregating all empirical phenomenon and discussions in a single piece, and given that the updated version of the paper includes some new analysis not found in prior work (Figure 4, and more analysis that is mostly in Appendix), I am fine with accepting this paper.

---

> ### Author Response · Authors · 2022-07-27
> **Overall response to Review G7uJ**
>
> Thank you for the detailed review and the extensive feedback! We are encouraged that you found the paper clear and coherent, and that it discussed a range of broader applications. We added substantial new content based on your suggestions. They are summarized here:
>
> 1. Overview of types of BIG-Bench tasks that emergent abilities occur (Appendix A.3)
> 1. Overview of types of MMLU supercategories that emergent abilities occur (Appendix B)
> 1. Extended discussion of risks, including emergent risks (Sec 5.4)
> 1. List of ~40 potential tasks that are candidates for future emergence (Appendix E.4)
> 1. New section discussing potential future directions since simply scaling more is challenging (Sec 5.6)
> 1. Plots with multiple metrics (ROUGE, BLUERT, etc) for three generative BIG-Bench tasks (Appendix A.2)
> 1. Empirical evidence on factors for emergent abilities beyond scale (PaLM 62B beats models that have more parameters and training compute) (Sec 5.2)
> 1. Up-weighted pointer to analyses with different axes of scale, such as model parameters (Figs 4, 10-12) (Sec 2 Par 2)
> 1. Softened language around further scaling potentially enabling new emergent abilities (Abstract, Intro)
>
> This paper was motivated by emergent abilities of LMs in the recent literature, and we hoped that aggregating them in a survey would motivate further research, as in line with the TLMR call for surveys (see https://www.jmlr.org/tmlr/faq.html). Our goal was not to run new experiments or to provide novel analysis. Although some plots aggregate previously reported results, after revising, now the paper contains seven figures that are not found directly in prior literature (e.g., Figs 4-10, Table 1).
>
> The detailed description of improvements we made is below, and if you have further suggestions for improving the paper please let us know and we will revise the paper more!

---

> > ### Author Response · Authors · 2022-07-27
> > **Detailed response to Review G7uJ (1/n)**
> >
> > > A broader overview on when emergent abilities occur / do not occur (e.g., in terms of tasks, models) that is beyond what has been already reported in prior work
> >
> > Thanks for this nice suggestion. We agree that it is interesting and natural to ask which kinds of tasks tend to have emergence. In the revision, we added in two new sections that attempt to address this question. They are Appendix A.3 and Appendix B, which we describe below.
> >
> > In Appendix A.3, we asked the question of whether certain categories of BIG-Bench tasks were more likely to be emergent. We manually labeled all 210 BIG-Bench tasks as (thus far) emergent or not on LaMDA, GPT, or PaLM (see paper for full details) and then considered what percentage of tasks associated with certain keywords (e.g., “commonsense”, “multilingual”) were emergent. The keywords were tagged by the BIG-Bench task authors. This is summarized in Figure 8. Though some reasoning categories had the largest share of tasks with emergent behavior, there were no obvious trends for which keywords had the largest share of emergent tasks, and the analysis is noisy because most keywords had less than twenty associated tasks. Overall, this analysis opens the door to future research in this direction.
> >
> > The second analysis we did for which types of tasks are more likely to be emergent is in Appendix B. This was similar to the BIG-Bench analysis above, but for MMLU instead. We stratified Gopher and Chinchilla’s emergence on MMLU by the four MMLU categories (social science, humanities, STEM, and other). The plots show that the social science and humanities categories had the largest jump in performance from the second-largest to the largest model, and STEM had the smallest jump in performance.
> >
> > While we intended for the primary contribution of the paper to be a survey on prior work and discussion around how to view emergence, we hope these additional analyses further strengthen the paper.
> >
> > > More systematic analysis on how exactly emergent abilities occur … hypotheses & empirical evidence on why emergent abilities occur, i.e., what leads to discontinuous improvements as scale increases
> >
> > Thank you for this comment. We agree that how/why emergent abilities occur is an important and valuable research question. This is an open question for future work that we consider out of scope of this paper, though we have done some preliminary analysis on whether metrics reveal how and why they occur (see Appendix A). This preliminary analysis revealed that cross-entropy loss of models does improve even if such improvements aren’t reflected in downstream metrics such as accuracy or exact match, though it was not possible to predict the model scale at which emergence would occur from this analysis. Moreover, in Section 5.2 we have cited related work in the literature that discuss why emergent few-shot prompting occurs (e.g., Xie et al., 2022; Chan et al. 2022; Zhang et al., 2021).
> >
> > Overall, the primary contribution of this survey is to highlight and consolidate an existing trend in various current literature, with the hopes of motivating future empirical work on this question of how/why emergent abilities occur. We intended for this to be in line with the survey paper criteria from TMLR (https://www.jmlr.org/tmlr/faq.html): “Ideally, we want survey papers that draw new, previously unreported connections between several pieces of work in an area, and/or that clearly highlight trends in the area and/or suggest currently open problems.”

---

> > > ### Author Response · Authors · 2022-07-27
> > > **Detailed response to Review G7uJ (2/n)**
> > >
> > > > More systematic survey & empirical evidence on the impact of evaluation metrics in evaluating emergent abilities, beyond what has been already reported in prior work (could be elaboration from current Section 5.1/5.3)... potentially with how different metrics change the result reported in Section 4
> > >
> > > Thanks for the comment and suggestion to add different metrics for the plots in Section 4. We added a new section in Appendix A.2 with the requested plots showing how different metrics may portray emergence differently. The summary of this is below.
> > >
> > > In Section 5.1 and the accompanying Appendix A, we analyzed six emergent BIG-Bench tasks.
> > > Because evaluation metrics like exact match and accuracy do not measure partial credit, we asked whether using cross-entropy loss as the evaluation metric explains emergence. The analysis suggested that while small models improve in some ways that EM/Acc do not capture, these tasks are still emergent and not predictable solely from the behavior of small models.
> > >
> > > In the revised paper, we also added Appendix A2, which shows many metrics for three generative tasks in BIG-Bench. This new plot shows all the other applicable metrics from BIG-Bench (BLEU, ROUGE, BLEURT, Sequence F1, Multiple choice grade). In all cases, the trends still appeared emergent (though some strict evaluation metrics were always zero). This further suggests that emergent abilities cannot be entirely explained by the use of evaluation metrics that do not give partial credit.
> > >
> > > > More systematic survey & empirical evidence on factors for emergent abilities beyond scale (could be elaborated from current Section 5.2)
> > >
> > > Thank you for asking us to add more empirical evidence on factors for emergent abilities beyond scale. We agree that it is important to show how scale is not the only thing that unlocks emergent abilities, and that other factors such as dataset quality or architectural differences may play a significant role. We added more discussion in Section 5.2.
> > >
> > > Noting the strong performance of PaLM 62B demonstrated in Chowdhery et al., we hypothesized that there might be some BIG-Bench tasks for which PaLM 62B demonstrated emergent behavior that was not present in LaMDA 137B or GPT-3 175B. We manually examined the scaling plots of all BIG-Bench tasks and found 14 tasks for which LaMDA or GPT-3 models had near-random performance, but PaLM 62B performed substantially above random. Since PaLM 62B has fewer parameters and less training compute than both LaMDA 137B and GPT-3 175B, these 16 scaling plots are empirical evidence that emergence depends on factors beyond scale. There are many differences between PaLM and LaMDA/GPT-3 and hence no definitive single factor explaining why PaLM 62B performs better, though one might hypothesize that major factors include higher-quality training data (e.g., more multilingual and code data) and architectural changes (e.g., split digit encodings).
> > >
> > > > More fundamental evidence on the claim "further scaling will likely endow even-larger language models with new emergent abilities" with more concrete discussion (with possibly evidence) on [1] *how those new emergent abilities would look like* and [2] *how further scaling will be possibly in a approachable way*.
> > >
> > > Thank you for asking for more discussion on examples that could make future emergence more concrete. We mentioned in the first paragraph of Section 5 that there are dozens of BIG-Bench tasks for which no few-shot prompted models (including PaLM) have achieved better than random performance. To make this more concrete, in the revised paper we added Appendix E, which lists these 45 BIG-Bench tasks. These tasks are potential candidates for future emergence if a more-capable model in coming years could achieve better-than-random performance on them in a few-shot setting.
> > >
> > > We also added several paragraphs in Section 5.6 discussing potential future research directions for training better models other than simply scaling, which is difficult due to computational cost and hardware and optimization challenges. These directions could include improved model architectures and training, data quality and scaling, and better prompting techniques (see the revised paper for references to nascent work in these directions).
> > >
> > > Finally, we softened the language of the sentence from “further scaling will likely endow” to “This raises the question of whether further scaling could potentially endow even-larger language models with new emergent abilities.” We also softened the claim in the abstract and introduction. Thank you for this feedback and let us know if you think this is not reasonable!

---

> > > > ### Author Response · Authors · 2022-07-27
> > > > **Detailed response to Review G7uJ (3/n)**
> > > >
> > > > > Section 5.3 briefly discusses emergent risks, focusing on security issues in extraction of training data. The paper should include (self-contained) discussion on broader risks in applying large language models, including models having bias/harmful contents.
> > > >
> > > > (The below text is the same as for Reviewer Jq3i, who had the same suggestion.)
> > > >
> > > > Thank you for asking us to further discuss emergent risks. In addition to memorization risks that we discussed in the initial submitted version, we also added discussion on how truthfulness, bias, and toxicity risks are affected by model size. This added discussion is in Section 5.4. These risks are not necessarily “emergent” in the precise definition from Section 2 since such risks are likely present to some degree in models of all sizes, but we agree that they deserve extended discussion. In short, the new content we added to the revised paper summarizes results presented from recent papers on risks of large language models.
> > > >
> > > > We also noted that there are also emergent risks that have not yet been characterized. Such risks could be in domains such as backdoor vulnerabilities, inadvertent deception, or synthesizing harmful content, and are discussed in detail in Hendrycks et al. 2021c (which we added a pointer to in the paper).
> > > >
> > > > > For instance, the paper could have included results or analysis on different axes of scale (e.g. model parameters, pre-training data size, etc, apart from FLOP)
> > > >
> > > > Thanks for this comment. We agree that it is good to analyze emergent abilities using different axes of scale. Hence, we have all the plots in Section 3 and 4 with model parameters as the x-axis in the Appendix, as alternate figures. In the revised paper we upweighed the pointer to this in Section 2 and in the captions for Figure 2 and 3, but if you have any suggestions for how we can make it even more clear, please let us know.

---

### Review · Reviewer_Jq3i · 2022-07-12

**Summary Of Contributions:**

This paper surveys the literature to study emergent abilities of large language models. The authors share a clear definition of emergence and illustrate it with several examples from the literature. They survey a range of prior work in order to categorize different types of emergent abilities. The paper highlights how emergence motivates future research on why such abilities are acquired and whether more scaling will lead to further emergent abilities. The survey is comprehensive, well-organized, and easy to follow. I really like this paper and am confident this paper would encourage greater conversation on this topic but also have some minor concerns/suggestions listed below.



**Requested Changes:**

Other than the weaknesses listed above, one thing I would like to see more on is some discussion on why these abilities emerge, which could also include a summary of the work coming from computational neuroscience and psychology, as well as theory (are there certain kinds of architectures that have created these new abilities for instance?)

**Strengths And Weaknesses:**

### Strengths:
- This paper touches on an important and timely topic.
- The survey is very comprehensive, well organized and well written.
- The insights offered in the survey could open up new lines of research.

### Weaknesses:
- The paper is currently focused on models trained on English data and could benefit from discussion of what emerging capabilities are emerging in these models versus multilingual models, and if there are trends that are similar or different.
- The section on emergent concerns is rather limited in discussion than the strengths, and I do acknowledge that there's a lot more work on risks than capabilities in this space, but the paper currently does not include discussion on what risks might emerge as more capabilities emerge.
- There's not much discussion on future work in the paper. What are some promising directions for future work?

---

> ### Author Response · Authors · 2022-07-27
> **Response to Review Jq3i**
>
> Thank you for the encouraging review and the thoughtful feedback! Our paper aimed to aggregate existing emergent abilities in the literature to help motivate future work, and we are glad you found the topic important and timely. Following your feedback, we added (1) a new subsection on promising directions for future work, (2) examples of multilingual emergence, (3) extended discussion of emergent risks, and (4) additional discussion of why emergence occurs, with references from theory and the computational psycholinguistics literature.
>
> Below is the detailed description of the additions we made to the paper based on your feedback. If there are any other improvements that can be made please let us know!
>
> > What are some promising directions for future work?
>
> Thanks for this suggestion to add some directions for future work. Given the scope of the paper, there is a wide range of potential future work. We added a new subsection in the paper to discuss this.
> As a short summary, some future work directions include the following:
>
> Further model scaling may enable new abilities (though it is computationally expensive and requires solving hardware challenges).
> Improved model architecture and scaling (e.g., mixture-of-experts, adaptive computation, memory-augmented models) could improve models beyond simply scaling.
> Data scaling is an orthogonal way to scale models that requires more data and training FLOPs but does not require more model parameters (e.g., Chinchilla).
> Better techniques for and understanding of prompting could lead to further emergent abilities (e.g, chain-of-thought prompting enables models to perform multi-step tasks, better understanding of models could lead to insights on how to elicit emergent abilities in smaller models).
> Frontier tasks: many tasks are still not doable by even the largest language models (e.g., dozens of tasks in BIG-Bench), and further work could explore why these abilities have not emerged and how to enable models to perform them.
>
> > The paper is currently focused on models trained on English data and could benefit from discussion of what emerging capabilities … in these models versus multilingual models.
>
> Indeed, extending language models to multilingual settings is an important direction for which exciting research is currently being done (e.g., XLM, mT5, BLOOM). In terms of emergence, we looked through the multilingual tasks in BIG-Bench and found two interesting results. First, in many cases, scaling up language models enables emergent performance (e.g., Hindi knowledge, Hindi toxicity, and Swahili–English proverbs for LaMDA). Second, sometimes emergence occurs for certain models but not others. E.g., for Hindi QA, LaMDA does not beat random while GPT-3 beats the average human rater; Persian QA emerges for PaLM 62B but does not emerge for LaMDA 137B. This is perhaps due to differences in training data (PaLM 62B uses 22% non-English data, whereas LaMDA only uses 10% non-English data).
>
> In the revised paper, we added multilinguality as a direction for more future work and mentioned this above result in the last paragraph of Subsection 5.6. Additionally, we added the Persian QA emergence result into Figure 2(D), replacing a prior plot on English figure of speech classification task.

---

> > ### Author Response · Authors · 2022-07-27
> > **Response continued**
> >
> > > What risks might emerge as more capabilities emerge?
> >
> > (The below text is the same as for Reviewer G7uj, which had the same suggestion.)
> >
> > Thank you for asking us to further discuss emergent risks. In addition to memorization risks that we discussed in the initial submitted version, we also added discussion on how truthfulness, bias, and toxicity risks are affected by model size. This added discussion is in Section 5.4. These risks are not necessarily “emergent” in the precise definition from Section 2 since such risks are likely present to some degree in models of all sizes, but we agree that they deserve extended discussion. In short, the new content we added to the revised paper summarizes results presented from recent papers on risks of large language models.
> >
> > We also noted that there are also emergent risks that have not yet been characterized. Such risks could be in domains such as backdoor vulnerabilities, inadvertent deception, or synthesizing harmful content, and are discussed in detail in Hendrycks et al. 2021c (which we added a pointer to in the paper).
> >
> > > I would like to see more … discussion on why these abilities emerge, which could also include a summary of the work coming from computational neuroscience and psychology, as well as theory (are there certain kinds of architectures that have created these new abilities for instance?)
> >
> > Thanks for the comment about adding in work from related disciplines. We alluded to some machine learning theory work in Section 5.2 and elaborated on it in the revised version. Although there has not yet been a definitive explanation for how few-shot prompting works, some theory work has tried to understand how a general language modeling objective contributes to downstream behavior (Wei et al., 2021a; Saunshi et al., 2021).
> >
> > We also added in the following points:
> >
> > - With scale held constant, certain data properties can enable few-shot prompting (though these results are not on the specific models/tasks that we evaluate). For instance, Xie et al., 2021 showed that few-shot learning emerges when pretraining documents have long-range coherence, which requires the LM to infer a latent document-level concept to generate coherent next tokens. In their experimental setting, few-shot learning emerged for both transformers and LSTM architectures.
> > - In another paper, Chan et al., 2022 showed how in-context learning emerges due to burstiness and having large numbers of rare classes. In their setting, which involved image classification, transformers demonstrated in-context learning while Vanilla RNNs and LSTMs did not.
> > - Computational psycholinguistics is another field that emergent behavior has been studied. This direction typically focuses on “small” neural networks in the context of syntactic rule-learning (e.g., English subject–verb agreement). For instance, Figure 5 in Zhang et al. 2021 shows a clear emergent “aha” moment, where models acquire the ability for subject-verb agreement at a particular threshold amount of training data when model parameters is fixed. A similar striking learning plot is shown in Abend et al., 2017.
> >
> > Overall, these papers hint that many variables beyond scale contribute to emergence. Though these past studies are on smaller models and different tasks than the ones we describe in the paper (pre-training interventions are expensive on large models), it seems likely that certain data properties or example frequencies unlock emergent behavior.

---

> > > ### Comment · Reviewer_Jq3i · 2022-08-07
> > > **Appreciate the thoughtful and comprehensive response**
> > >
> > > Thank you for taking the time to write this detailed response. I am thrilled to recommend acceptance for this work and am very much looking forward to the discussion it enables in the community.

---

### Review · Reviewer_4m8p · 2022-07-17

**Summary Of Contributions:**

This work proposes the concept of Emergence to explain a phenomenon of large language models that certain abilities only emerge when the models are large enough. In this paper, it defines an ability is emergent if it is not present in smaller models but is present in larger models. Instead of using model size or other factors, this work suggests using training compute (FLOPs) to measure the model size. The concept of Emergence was tested with five large language models with eight categories of tasks.

**Requested Changes:**

Further changes are expected to address the aforementioned limitations. Particularly,

- A comparison of model performance based on model size instead of training FLOPs with the same training resources
- Additional performance tests, besides the prompt-bassed ones.

**Strengths And Weaknesses:**

**Strengths**

- The paper is well-written and easy to understand.
- The proposed concept of Emergence is interesting and as expected in this work, which may simulate future work on understanding and explaining the learning behaviors of large language models.
- The analysis across multiple large language models and several different tasks seems solid.

**Weaknesses**

- The results in figure 2 are probably misleading. For any given model with fixed numbers of parameters, smaller training FLOPs could mean the models are not well trained, instead of considering them as smaller models. To my understanding, drawing conclusions from training FLOPs could be the main limitation of this work. A piece of more convincing evidence could be a comparison across different models with varying model parameters trained on the same resources. If Emergence is really a property associated with model size, it is reasonable to expect its appearance regardless of model architectures (and other factors related to how to build a model).
- As explained in the paper, model performance has too many confounding factors. However, the argument of mainly using FLOPs is not convincing. More experiments and analyses like figure 4 are needed in this work.
- In addition to the prompt-based test, I am wondering whether the performance can also be tested with other techniques.

---

> ### Author Response · Authors · 2022-07-27
> **Response to Review 4m8p**
>
> Thank you for the insightful review! The goal of this paper was to survey recent emergent abilities found in the literature in order to stimulate future work on understanding and unlocking such behaviors, and we are glad you found the paper well-written and interesting. Following your suggestions, we (1) added extended analysis on Figure 4 (MMLU/different x-axes) in Appendix B, (2) added more on performance beyond prompting such as finetuning and calibration, and (3) gave more weight to pointers to plots with varying model size (Figs 10 and 11).
>
> Below is a detailed description of the additions we made to the paper. Please let us know if you have any further suggestions on how to improve the paper!
>
> > More experiments and analyses like figure 4 are needed in this work.
>
> Thank you for asking for more analyses like Figure 4. We agree that it is interesting to compare Gopher and Chinchilla models, since they have the same training FLOPs and are evaluated on the same Wikitext103 and MMLU datasets. As further analysis, we asked the question of whether certain MMLU tasks were more emergent than others. We added an Appendix Section A.3, which stratifies the MMLU results into four categories (Humanities, STEM, Social Science, and other) with three x-axes (training FLOPs, model params, and WikiText ppl). This stratification revealed that the most emergent categories were humanities and social science. The least emergent category was STEM, and the four least emergent sub-topics were also all STEM (high school math, abstract algebra, college math, and high school physics). This result suggests that language models still struggle with tasks that require multi-step abstract reasoning, which may be a direction for emergence in future language models. We also added a plot that summarizes this result (Figure 8).
>
> > In addition to the prompt-based test, I am wondering whether the performance can also be tested with other techniques.
>
> Thank you for this interesting comment. In addition to prompting, another way of using language models is finetuning on a specific task—this has worked well in the case of “Scratchpads for intermediate computation”, in which Nye et al. 2021 showed that the ability for a finetuned model to do 8-digit and 9-digit addition emerged with sufficient model scale. This is shown in Figure 3C. Figure 3B is also related in that it uses multi-task finetuning (though it is followed by zero-shot prompting).
>
> We also looked further for additional use cases, and another way of evaluating language models is calibration, from the very recent paper “Language models (mostly) know what they know” by Kadavath et al., 2022. This paper showed that directly asking large language model whether an answer was true/false can outperform standard calibration methods (e.g., measuring the probability of the correct answer among several options). The superiority of this true/false method only emerges when scaled to the largest 52B language model. We added this in as an additional plot in Figure 3D.
>
> If we have missed any examples of emergent behavior in scenarios beyond prompting please let us know and we will add it in!

---

> > ### Author Response · Authors · 2022-07-27
> > **Response continued**
> >
> > > For any given model with fixed numbers of parameters, smaller training FLOPs could mean the models are not well trained, instead of considering them as smaller models… a piece of more convincing evidence could be a comparison across different models with varying model parameters trained on the same resources
> >
> > Thanks for highlighting the point that many factors probably contribute to emergence. Indeed, there are no perfectly controlled comparisons ablating each factor due to the high computational cost of pre-training large LMs. So far, LMs have primarily been scaled on three axes: training FLOPs, model parameters, and data size. Because training FLOPs and model parameters are typically increased simultaneously, we plotted the Section 3 and 4 figures (Figs 2 and 3) in both training FLOPs and model parameters. The plots with varying model params are Figures 10 and 11. In the revised paper, we gave further emphasis to the pointer to these plots with varying model parameters (Section 2 Paragraph 3).
> >
> > Chinchilla is a recent paper that is perhaps closest to your comment. Chinchilla has the same training FLOPs as Gopher, but decreases model parameters while increasing data size. The better performance of Chinchilla is one motivation for showing Figure 2 with the x-axis as training FLOPs, and the figure with x-axis as model parameters in the Appendix. We agree that it is wise to view emergence as a function of many variables (e.g., Figure 4, as you saw, has training FLOPs, model parameters, and Wikipedia perplexity).
> >
> > For Figure 2, we tried to clarify that each point on the plot is a different model and not the same model at intermediate training steps by saying “model scale” in the x-axis label and “each point is a separate model” in the caption. While results for intermediate checkpoints of large LMs are typically not reported, we think that they could be misleading points on this plot if they were. We hope that Figure 10, which is a replica of Figure 2 with model parameters on the x-axis, would additionally clarify that models typically increase parameters and training FLOPs simultaneously. Let us know if we can make it more clear!
> >
> > Finally, it is interesting to speculate why existing work has not explored the setting where both training FLOPs and data scale are held exactly constant, and model parameters varies. One challenge in this setting would be that either the small models would be over-trained or the large models would be under-trained. So this setting would either be biased against small or against large models, making the comparison unfair. The expensive cost of training large models also makes it hard to justify spending the same amount of FLOPs for a 500B model as a 1B model, for example.

---

### Author Response · Authors · 2022-07-27
**Thank you for the reviews**

We thank all three reviewers for the comprehensive suggestions. To summarize, our paper surveys recent literature on emergent abilities of large LMs and discusses how to view emergence, with the goal of motivating future work investigating this phenomenon. To the best of our knowledge, this is the first survey to aggregate emergent abilities of large LMs in a single place.

Reviews 4m8p and Jq3i mentioned strengths such as “important” and “timely” and thought the survey would inspire future work; they also requested changes that we added in the revision.

Review G7uJ found strengths in the paper’s clarity and broader implications, and had a longer list of suggestions for improvement. As described in detail in the G7uJ response, we added significant additional content to address these suggestions, and the revised paper now includes many additional analyses not found in prior work (e.g., Figures 4-10).

The intention is for the paper’s primary contributions to be (1) a consolidation of various settings where emergent abilities have been previously observed, (2) discussion around how emergence might be viewed, and (3) motivation for future work characterizing why/how emergence occurs and how to enable emergence beyond simply scaling. Note that TMLR calls for surveys specifically: “Ideally, we want survey papers that draw new, previously unreported connections between several pieces of work in an area, and/or that clearly highlight trends in the area and/or suggest currently open problems.” (https://www.jmlr.org/tmlr/faq.html). We believe this paper’s contributions list above are in line with these criteria.

In summary, the revisions include 1.5 additional pages in the main paper and 4 new pages of Appendix content:
- [4m8p, G7uJ] Overview of Chinchilla / Gopher’s emergence on MMLU in terms of what supercategories emergence occurs for, as an extension of Fig 4 (Appendix B)
- [4m8p] Emergent calibration result as an example in addition to prompting (Fig 3D)
- [Jq3i, G7uJ] Extended discussion of risks, including emergent risks (Sec 5.4)
- [Jq3i, G7uJ] New section discussing potential future directions since simply scaling more is challenging (Sec 5.6)
- [Jq3i] Discussion of multilingual emergence (Sec 5.6) with an example in Fig 2D
- [Jq3i] Discussion on why abilities emerge, with references from theory and computational psychology (Sec 5.2)
- [G7uJ] Overview of when emergent abilities occur / do not occur in terms of tasks (Appendix A.3)
- [G7uJ] Plots with multiple metrics (ROUGE, BLUERT, etc) for three generative BIG-Bench tasks (Appendix A.2)
- [G7uJ] Empirical evidence on factors for emergent abilities beyond scale (PaLM 62B beats models that have more parameters and training compute) (Sec 5.2)
- [G7uJ] List of ~40 potential BIG-Bench tasks that are candidates for future emergence (Appendix E.4)
- [4m8p, G7uJ] Up-weighted pointer (Sec 2, Par 2) to analyses with different axes of scale, such as model parameters (Figs 4, 10-12)
- [G7uJ] Softened language around “further scaling potentially enabling new emergent abilities” (Abstract, Intro)

Thanks again for the review process! If there is anything we have missed or if there are further ways we can improve the paper please let us know and we will be happy to work on it.

---

### Public Comment · ~Joshua_Kimrey1 · 2023-03-31
**Potential Data Misrepresentation**

It can be very hard to discern the functional relationship between two variables in a scatter plot when one of the axes is log-scaled and the other axis is linear scaled. This configuration of axes scaling is used in all the panels of Figure 2; all of the horizontal axes are in log-scale while all of the vertical axes are in linear scale. As the central claims of the paper are deduced from these kinds of visual representations of the relationships between compute resources/parameters and model accuracy, can the authors produce plots of these same data with both axes on the same, linear, scale? I am asking because, for example, the configuration of mismatched axes scaling that is used throughout the paper would make even linear growth appear visually as exponential growth. The claims of "emergence" rest on the displayed relationships being more than artifacts of expedient plotting choices.

---

> ### Author Response · Authors · 2023-03-31
> **Emergence is not dependent on whether the x-axis scale is linear or log**
>
> I don't think many people find this data to be misrepresented, but I can understand where you are coming from. Does this help clarify?
> 1. In practice, models are scaled up exponentially, not linearly. For instance, PaLM goes from 8B to 62B to 540B. And so the exponential x-axis is the best representation how language models are improving. In the literature the x-axis will typically be log-scale, otherwise you have a cluster of points at the bottom-left and not anywhere else. For instance, see the plots in the GPT-3 paper.
> 2. Even the x-axis is abstracted away, the point is that there is a qualitatively different behavior from one model to the next-biggest one still stands.
> 3. Even if you find the above points unconvincing, it's still possible to see emergence on a linear x-axis scale. However, I don't find this representative of how language models are scaled and don't think it's a productive use of time for readers. I plotted Figure 2A [here](https://docs.google.com/spreadsheets/d/1uWAtODZmmzhKxDrBXJqjufEtEu56PzfhhV78lGcl1b4/edit?usp=sharing), and you'll still see the same emergent spike, albeit in a less readable way. (If you want all of them, just download the LaTeX source or find the data points in the corresponding papers.) And just to clarify, the visualization of the plot is independent of the data, and the original point still holds: emergent behavior cannot be predicted simply by extrapolating the scaling curve of smaller models.
>
> As an aside, if you think there is some way to predict emergent behavior in larger models, would love to hear more and read / champion your work if you write it up. As I have said on twitter, dinner on me at the next conference if you write a paper about emergence: https://twitter.com/_jasonwei/status/1622763322562199552
>
> -Jason Wei

---

> > ### Public Comment · ~Joshua_Kimrey1 · 2023-04-01
> > **What are we calling emergence then?**
> >
> > Thank you for replying and for taking the time to produce a re-scaled plot.
> >
> > To your point 1.: how model sizes scale in practice should be immaterial to how one estimates the functional relationship between compute and accuracy. How can one be confident that there is "emergent behavior" at a threshold compute/parameter value if accuracy results are missing for almost every would-be model of intermediate size in the depicted interval? With the data presented we cannot know for sure that the accuracy for models of intermediate size does not simply interpolate linearly between the accuracy results of the second largest model and the accuracy results of the largest model.
> >
> > To your point 2.: my question from point 1. stands. You are comparing the results of two models which are of different orders of magnitude in size. Quantity may have a quality all its own.
> >
> > To your point 3.: visually, it is unclear to me that the plot you re-produced gives indications that the accuracy observed for the largest model "cannot be predicted simply by extrapolating the performance of smaller models". If anything, it appears that linear extrapolation from the smaller models might have led one to expect a higher accuracy than is observed for the largest model displayed.
> >
> > Before we start trying to explain or predict "emergent behavior", should we not first make sure that it actually exists where we claim it does? Or at least define what we mean in a way that makes it more amenable to scientific falsification?

---

> > > ### Author Response · Authors · 2023-04-01
> > > **The emergence in the plot occurs at 7B and not 175B**
> > >
> > > Points 1 & 2. Yes, it's unfortunate that we don't have those intermediate model sizes, and I wish that we did. However, there must be model threshold at which performance moves from random to non-random, and there are several model sizes where performance is totally random. And for emergent prompting techniques, there must be some point when the performance delta from the prompting technique becomes positive. So I think emergence would still exist, even if you imagined the intermediate model sizes in the plot.
> > >
> > > Point 3. The emergence in this plot is not for the largest model but rather at the second-largest model (7B), which would not be predictable by the smaller models in my view.
> > >
> > > Point 4. Yes, it is a good point to have a more concrete definition. My personal opinion was that I did not find this to be super productive. You might say something like "an ability emerges when it is statistically better than random guessing" or "an ability emerges when it is 5% better than random guessing", but the former depends on the eval dataset size and the latter has an arbitrary threshold.
> > >
> > > I don't know what it would mean for an emergent ability to be falsified, but there are plenty of abilities that are not emergent. For example, many abilities arise smoothly as a result of model scale, some abilities are inversely correlated with model scale, and yet other abilities never achieve performance above random for any model.
> > >
> > > If it's any consolation prize, Figure 5 has error rate plotted on the log scale, and it is still emergent. You can also plot cross-entropy of the targets, for which we found that cross-entropy improves for small models but there is still an unpredictable spike when emergence occurs (full discussion in Appendix A).
> > >
> > > Hope this helps,
> > > Jason

---

> > > > ### Public Comment · ~Joshua_Kimrey1 · 2023-04-12
> > > > **Risk of doublespeak?**
> > > >
> > > > My issue is with statements like: "For example, many abilities arise smoothly as a result of model scale,...". You have discrete data, a very small number of data points, and each value of FLOPs/parameter count for which there is data differs from its neighboring data points by roughly an order of magnitude. Any discussion of whether the relationship between accuracy and FLOPs/parameter count is smooth or that they are correlated can only be conjecture. There simply isn't enough data to confirm or deny these claims.
> > > >
> > > > Do you mean to imply that the existence of "emergence" excludes the possibility for smooth interpolation? What do you mean by smooth? Do you mean something other than the requirement for a continuous derivative?
> > > >
> > > > Can you exclude the possibility that these models are improving in accuracy at a rate that is roughly linear in parameter count, but that the early improvements are obfuscated by the inherent stochasticity of the optimization problem? As in, what if it is the case that the median accuracy for a model of a fixed size is much smaller than the natural variability in accuracy over different model realizations at that size? This would make the models appear to be no better than random until they reached an accuracy that was sufficiently larger in magnitude than the natural variability in model performance.
> > > >
> > > > If there is no definition of what constitutes an "emergent ability", how can the claims made in this and other works be subject to further scientific scrutiny? How can I tell whether I am observing an emergent ability or something that is simply a surprising model behavior?

---

> > ### Public Comment · ~Gary_Smith2 · 2023-04-12
> > **raw data**
> >
> > Hi Jason,
> >
> > Where is the link to the raw data?
> >
> > Thanks in advance

---

### Decision · Action_Editors · 2022-08-17

**Recommendation:** Accept as is

**Comment:**

This paper discusses the phenomenon of special abilities emerging in large language models that are not present in smaller models and cannot be easily predicted using the performance of the smaller models. The authors present a comprehensive survey of existing results showcasing such abilities, along with a detailed discussion of broader implications including potential sociological risks, emergence in smaller models and future applications. Based on reviewer feedback, the authors have revised the paper to include new insightful analyses that go beyond aggregating existing results in literature. Overall, this paper provides a valuable survey and discussion that will inform future research involving large LMs.